# Magnetosensitivity of tightly bound radical pairs in cryptochrome is enabled by the quantum Zeno effect

Matt C. J. Denton [1,2], Luke D. Smith [1,2], Wenhao Xu [1,2], Jodeci Pugsley [1,2], Amelia Toghill [1,2] & Daniel R. Kattnig [1,2] ✉

The radical pair mechanism accounts for the magnetic field sensitivity of a large class of chemical reactions and is hypothesised to underpin numerous magnetosensitive traits in biology, including the avian compass. Traditionally, magnetic field sensitivity in this mechanism is attributed to radical pairs with weakly interacting, well-separated electrons; closely bound pairs were considered unresponsive to weak fields due to arrested spin dynamics. In this study, we challenge this view by examining the FAD-superoxide radical pair within cryptochrome, a protein hypothesised to function as a biological magnetosensor. Contrary to expectations, we find that this tightly bound radical pair can respond to Earth-strength magnetic fields, provided that the recombination reaction is strongly asymmetric—a scenario invoking the quantum Zeno effect. These findings present a plausible mechanism for weak magnetic field effects in biology, suggesting that even closely associated radical pairs, like those involving superoxide, may play a role in magnetic sensing.

Magnetoreception is widespread within the animal kingdom and a central aid to the navigation of migratory animals[1,2]. Despite being actively researched for the past half century[3], the mechanistic underpinnings of this fascinating sense remain elusive. For several species, such as migratory songbirds, circumstantial evidence suggests that magnetoreception could be facilitated by the spin dynamics of radical intermediates within the flavoprotein cryptochrome[4,5]. However, beyond the conceptual picture[6,7], many questions remain unanswered, not least surrounding the identity of the magnetosensitive radical pair that the protein is presumed to accommodate[8,9].

A radical pair formed in cryptochrome through a photo-induced electron transfer process from a tryptophan (W) residue to the flavin adenine dinucleotide (FAD) co-factor, i.e. $FAD^{\bullet-}/W^{\bullet+}$, has been thoroughly investigated[10–15]. Despite evidence in favour of this system, such as sensitivity to the intensity of moderately strong magnetic fields in vitro, its role in magnetoreception is not uncontested. Several studies argue that magnetoreception may not be directly linked to the photo-

excitation of cryptochrome with short-wavelength light, a prerequisite for the generation of such a radical pair[16–18]. In addition, doubts persist as to whether $FAD^{\bullet-}/W^{\bullet+}$ could truly deliver the sensitivity required for a viable magnetic compass in the weak geomagnetic field ($50\,\mu T$), as the predicted directional magnetosensitivity of this system is strongly impaired by the hyperfine interactions spread about both radicals[19–21], spin relaxation[22,23] and inter-radical interactions[24].

Another proposed radical pair arises from the light-independent reoxidation of the (photo-)reduced cryptochrome with molecular oxygen[9,16,25,26], motivated by experimental[27,28] and previous theoretical work[29–31]. This process gives rise to the FAD semiquinone (FADH$^{\bullet}$) and the one-electron reduced form of oxygen, the superoxide anion ($O_2^{\bullet-}$), forming the radical pair FADH$^{\bullet}$/$O_2^{\bullet-}$, initialised in the spin triplet (T) state[32]. With a nuclear spin quantum number of $I = 0$ for oxygen, $O_2^{\bullet-}$ could in principle be advantageous for the purposes of magnetoreception[33], as having all hyperfine-coupled nuclear spins situated on one of the radicals greatly enhances magnetic field effects

[1]Living Systems Institute, University of Exeter, Stocker Road, Exeter, Devon EX4 4QD, UK. [2]Department of Physics, University of Exeter, Stocker Rd, Exeter, Devon EX4 4QL, UK. ✉e-mail: d.r.kattnig@exeter.ac.uk

(MFEs) at low magnetic field strengths[20,21,26]. Indeed, the idealised MFEs predicted for this system in the geomagnetic field exceed those of RPs such as a FAD$^{\bullet-}$/W$^{\bullet+}$ by at least one order of magnitude[22,34].

Notable limitations of the FADH$^{\bullet}$/O$_2^{\bullet-}$ radical pair are the deleterious effects of spin relaxation arising through the spin rotational mechanism[35–37], and of electron-electron dipolar (EED) coupling of the radical pairs at short inter-radical distances on the magnetosensitive dynamics[24,38,39]. While the latter aspect is widely acknowledged (see e.g.[8]), the former has not been considered in detail. Arguably, however, they are linked. When tumbling in solution, superoxide undergoes swift spin relaxation as its large spin-orbit coupling interaction brings about an efficient coupling of the spin motion to the molecule's total angular momentum, and thus its tumbling motion, which suppresses MFEs in weak magnetic fields[35,37]. Overcoming this obstacle requires immobilisation of the superoxide at the reaction distance, inducing large inter-radical coupling that is again expected to suppress magnetosensitivity[31,40]. A study by Mondal et al. has, for example, identified a possible immobilisation/reaction site of superoxide in the FAD binding pocket of cryptochrome[31,41]. Due to the comparatively short inter-radical separation of ~ 4.5 Å, the EED interaction of such a bound radical pair would invoke inter-radical couplings up to −1.7 GHz, thus exceeding the electron's Larmor precession frequency in the geomagnetic field (1.4 MHz) by a factor of order $10^3$ and typical hyperfine interactions by at least a factor of 40. As the eigenstates of the EED interaction are the singlet and triplet states, this leads to suppression of the coherent singlet-triplet interconversion. Fluctuations of the inter-radical interactions, arising from fluctuations in inter-radical distance, further induce decoherence effects that serve to curtail the evolution of the system, in turn reducing or eliminating MFEs[24,39,42]. Despite these issues, the FADH$^{\bullet}$/O$_2^{\bullet-}$ model has remained popular not only within the context of magnetoreception, but many biological processes for which a weak-field magnetosensitivity and/or magnetic isotope effects have been observed[43], such as neurogenesis[44] or cellular bioenergetics[45]. In addition, several models pair superoxide with alternative radical partners, such as a tryptophan radical cation[46,47]. For these, the complications associated with superoxide spin relaxation, immobilisation and strong inter-radical coupling as discussed here apply just the same.

The quantum Zeno effect, also referred to as the chemical Zeno effect[48], describes the retardation of state evolution brought about by fast, repeated measurements of a system[49,50], or more generally an arbitrary quantum operation or quantum semigroup[51]. The manifestation of quantum Zeno dynamics in this case is performed by virtue of the spin-selective recombination reaction of the radical pair. Previous investigation by Kominis and co-workers has demonstrated how the quantum Zeno effect arises in the regime of asymmetric recombination rates[52], and subsequently recognised the principal potential of this effect to counteract the sensitivity loss due to the exchange coupling and enhance the magnetosensitvity of simple radical pair systems subject to weak inter-radical coupling[53]. Recently, we have studied a model of a radical pair undergoing one-dimensional diffusive motion and found that its enhanced magnetosensitivity relative to statically immobilised RPs is, at least in part, attributed to the effect operative during encounter events[54]. By leveraging the quantum Zeno effect to counteract the sensitivity loss in the Radical Pair Mechanism (RPM), greater magnetosensitivity can be achieved in principle, i.e. in toy models. However, models specific to the directional magnetic field effects in cryptochromes in general and FADH$^{\bullet}$/O$_2^{\bullet-}$ in the cryptochrome binding pocket, i.e. with extreme EED coupling, in particular have not been explored. The activation of molecular oxygen by flavins and flavoproteins has been reviewed extensively[32,55], and we anticipate that the required strongly asymmetric recombination rates can arise in radical pairs such as these based on: estimates of the rates across reaction stages[32], on density functional calculations of the fast reaction of the flavin semiquinone with hydroperoxyl radicals[56,57], and on

investigations of the reoxidation of the reduced or semi-reduced flavin cofactor for plant cryptochrome AtCry1[27,28]. Whilst the reaction of FADH$^{\bullet}$ and superoxide/hydroperoxide in the binding pocket could not be resolved in these experiments, it is expected to occur at a fast rate[58]. We provide a detailed discussion on the reaction kinetics of the flavin semiquinone-superoxide reaction in the Supplementary Information (SI) and note that additional singlet-triplet dephasing can further relax the requirement of strongly asymmetric reactivity (see Suppl. Fig. 7).

Spin relaxation, and fluctuations of inter-radical interactions, in the context of the quantum Zeno effect is also unexplored for representative systems. Therefore, it was unclear if the quantum Zeno effect could counteract sensitivity loss in superoxide radical pairs in close contact, which suffer from strong and significantly fluctuating inter-radical coupling. Resolution of this is further complicated as the traditional method of treating spin relaxation, facilitated by Redfield theory, becomes inaccurate for situations where fast spin-selective recombination is operative[59,60].

Here we investigate the influence that the quantum Zeno effect could exert on a triplet-initialised flavin semiquinone-superoxide system. The archetypal energy level structure of flavin-semiquinone (as determined by the most pertinent hyperfine and EED interactions) is shown to be ideally suited to support directional MFEs in combination with the quantum Zeno effect. We explore systems of varying natural complexity to identify key factors and determine the robustness of the effect to spin system size, deviations from idealised geometries and involvement of spin relaxation. By using molecular dynamics (MD) simulations of superoxide bound in cryptochrome to inform our spin dynamics calculation, we identify regions in the relevant parameter space where magnetosensitivity is sustained or even enhanced compared to models excluding environmental interactions. To probe spin relaxation in the complex regime in which the quantum Zeno effect influences the dynamics, an approach based on the Nakajima-Zwanzig equations of motion in the Schrödinger picture[61,62], as suggested by Fay et al. was adopted to overcome the issues of the Redfield-based approach. Our study suggests a pathway for superoxide-based radical pairs, previously rejected due to strong inter-radical coupling and spin relaxation, to provide MFEs in biological processes. More broadly, immobilised, tightly bound and strongly coupled radical pairs that had previously been precluded as hosts of MFEs, could unexpectedly be magneto-sensitive nonetheless.

## Results

We base our investigation on the reaction scheme for a FADH$^{\bullet}$/O$_2^{\bullet-}$ radical pair shown in Fig. 1. Initially, the fully reduced form of FAD, FADH$^-$, reacts with molecular oxygen, transferring an electron to the oxygen molecule. As molecular oxygen has a triplet (T) ground state, this initialises the radical pair in the T-state. Subsequently, the radical pair coherently and incoherently evolves on a nanosecond timescale[7]. The pair's fate depends on its electronic spin state. Either the singlet state radical pair can bind O$_2^{\bullet-}$ to form a hydroperoxide (with associated rate constant $k_r$), the subsequent decomposition of which releases H$_2$O$_2$ and reoxidises the FAD cofactor; or the O$_2^{\bullet-}$ can escape the reaction site with rate constant $k_f$, independent of spin configuration. From this it follows that the overall rate constants of singlet and triplet depopulation, $k_S$ and $k_T$, respectively, are given by $k_S = k_r + k_f$ and $k_T = k_f$. In our simulations, we consider singlet and triplet reaction rates in the range of $10^{-3}\,\mu s^{-1}$ to $10^6\,\mu s^{-1}$, formally corresponding to lifetimes between 1 ms and 1 ps. The details of the simulation approach are laid out in the Methods section and the Supplementary Information. The essential features are that we consider coherent evolution due to the Zeeman, hyperfine and electron-electron dipolar interactions, include asymmetric recombination via the Haberkorn approach[63], and treat spin relaxation in the presence of asymmetric recombination based on the Nakajima-Zwanzig projection operator framework retaining terms to second order in the Markovian limit[60].

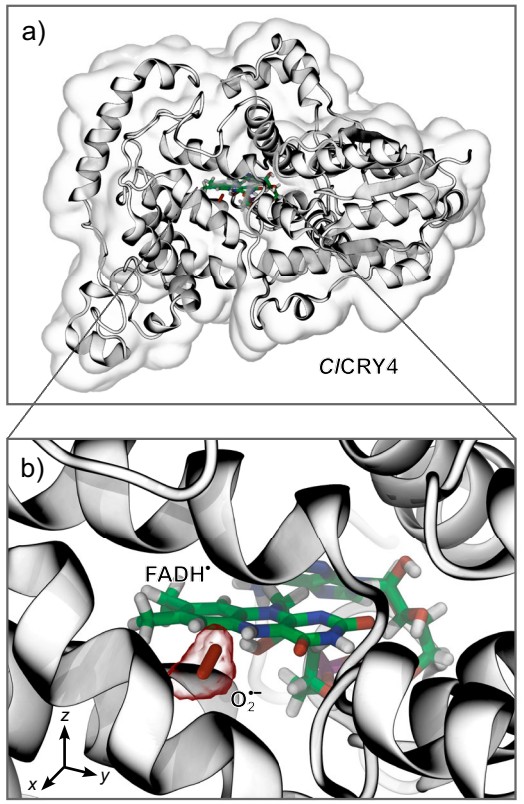

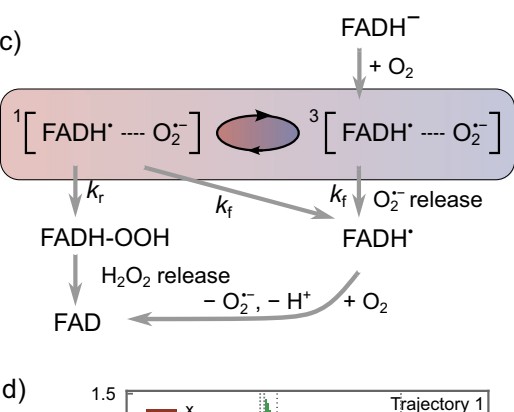

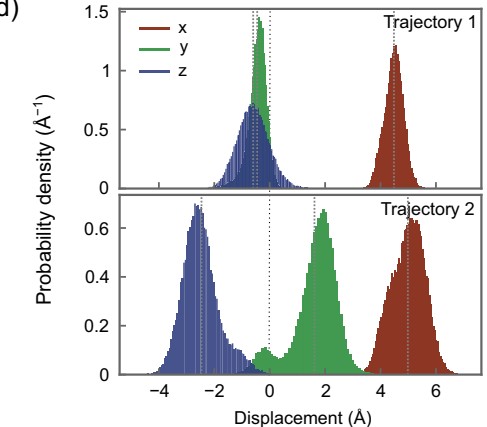

**Fig. 1 | Cryptochrome radical pair model and corresponding reaction scheme.**
**a** shows the pigeon (*Columba livia*) cryptochrome 4a (ClCRY4a; PDB ID: 6pu0), currently the only avian cryptochrome with a resolved structure[84], with the FADH$^\bullet$ and superoxide (O$_2^{\bullet-}$) molecules shown in colour, and in close-up in **b**. **c** shows the reaction scheme for the FADH$^\bullet$/O$_2^{\bullet-}$ radical pair, with the shaded box indicating where the magnetosensitive spin dynamics occurs. **d** shows the probability density

of the displacement of the centres of spin densities of the two radicals in the molecular frame of FAD for the two studied MD trajectories, highlighting the non-ideal placement of O$_2^{\bullet-}$ below the ring place in Trajectory 2. Dotted lines in figures b and c indicate mean values of each distribution. (Source Data are included as a source data file).

We validated this approach relative to the hierarchical equations of motion (HEOM) technique[64], and found it to apply without reservation for the considered scenarios (see Suppl. Fig. 16 and Suppl. Fig. 17 for details).

We commence by establishing the suppressive effect of EED interactions on magnetosensitive spin dynamics and its resolution for a toy model of FADH$^\bullet$/O$_2^{\bullet-}$, comprising only the most dominant hyperfine interaction associated with the N5 nucleus ($I = 1$). Assuming an idealised axial hyperfine coupling, Fig. 2 illustrates the distance dependence of the directional magnetic field effect by plotting the difference of the maximum and minimum singlet yields, $\Delta\Phi_S$, as obtained for 300 relative orientations of the inter-radical vector (evenly spaced over a Fibonacci sphere), as a function of the inter-radical distance $r$. For panel a), homogeneous recombination with $k_S=k_T=1\,\mu s^{-1}$ was assumed, while for panel b), $k_S = 2000\,\mu s^{-1} \gg k_T = 1\,\mu s^{-1}$. From a), it is apparent that the magnetosensitivity is suppressed for small $r$ regardless of the relative radical orientation, and that distances in excess of ~50 Å are necessary to realise the MFE of the pure hyperfine mechanism, unmitigated by the EED interaction. At larger distances, however, the spin-selective recombination reactions cannot possibly proceed at the prescribed rate constants, and thus these large effects are merely idealisations, unattainable under realistic conditions. At intermediate distances, a few resonant enhancements of $\Delta\Phi_S$ are observed when a component of the EED interaction matches the dominant hyperfine splitting $A_{ZZ}$. However, these only elicit comparably small MFEs and are specific to the simple system involving a single hyperfine-coupled nucleus; more complex systems show greater suppression of MFEs[24,65]. For FADH$^\bullet$/O$_2^{\bullet-}$, which reacts by bond formation

at the contact of the two radicals (~5 Å), these data imply that no significant magnetosensitivity is expected for $k_S = k_T$ even in idealised toy models based on the suppressive effect of the EED interaction. This situation changes remarkably if $k_S$ is increased. For $k_S = 2000\,\mu s^{-1} \gg k_T = 1\,\mu s^{-1}$ (panel b), we witness a remarkable sensitivity boost for small $r$ and equal or larger MFEs at greater $r$. We see that whilst MFEs are present at the low geomagnetic field considered in this study, it persists and the effect increases up to tens of mT (see Suppl. Fig. 10). For this one-nitrogen radical pair model we have also analysed the time dependence of the singlet and triplet probabilities (see Suppl. Fig. 11), for which the triplet probability shows a marked difference in $x$ and $z$ directions. Critically, the effects at small distance strongly depend on the relative orientation of the radicals, i.e. exceptionally large or absence of significant MFEs are observed depending on the relative radical placement. A detailed analysis (see Suppl. Fig. 1) reveals that large sensitivity ensues if, and only if, $r$ is perpendicular to the dominant hyperfine axis, and if the initial spin state is a triplet rather than singlet (see Suppl. Fig. 12).

In Fig. 2c we present the results of a systematic evaluation of the MFE as a function of the recombination rate constants $k_S$ and $k_T$ for superoxide displaced by 4.56 Å from the FADH$^\bullet$ along the $x$-axis (in the FAD molecular frame). The chosen inter-radical distance corresponds to the mean distance as observed in previous molecular dynamics (MD) simulations of superoxide in the FAD-binding pocket[31]; the displacement direction has been idealised to meet the optimality criterion from above. It is apparent that the region for which $k_S \gg k_T$ shows greatest potential for strong MFEs. Two branches of large effects are seen in the double-logarithmic plot, for which the optimal $k_S$ and $k_T$

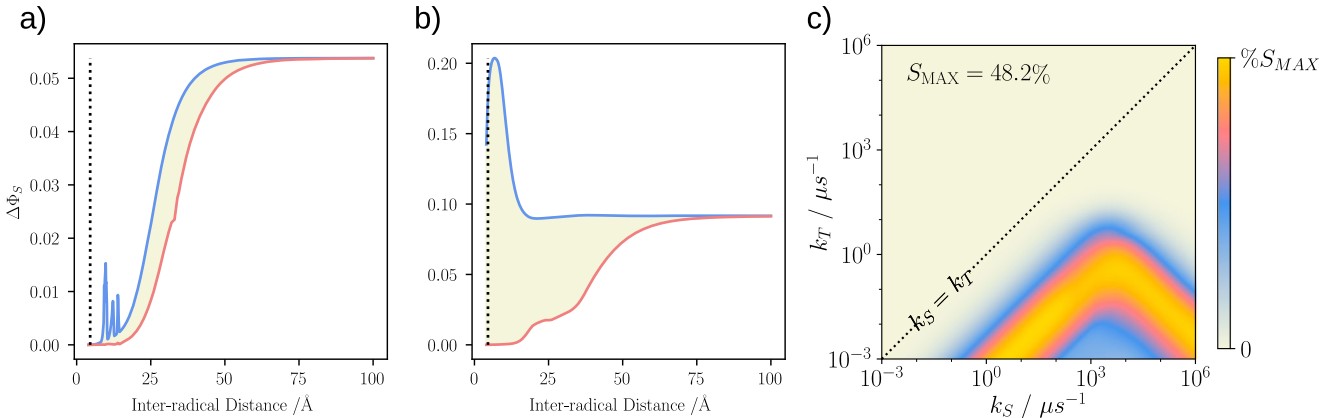

**Fig. 2 | Directional magnetic field effects of a one-nitrogen toy model of FADH·/O$_2^{·-}$.** Panels a) and b) show the minimum (red) and maximum (blue) singlet yield anisotropy, $\Delta\Phi_S$, varied over inter-radical orientations, as a function of the inter-radical distances for two different scenarios of spin-selective recombination, where the shaded region indicates the range of anisotropy values. **a** homogeneous recombination with $k_S = k_T = 1\,\mu s^{-1}$, and **b** strongly asymmetric recombination with $k_S = 2000\,\mu s^{-1}$, $k_T = 1\,\mu s^{-1}$. The black dotted line indicates an inter-radical distance of 4.56 Å, typical for a FADH·/O$_2^{·-}$ radical pair. At this separation, asymmetrical recombination rates result in strongly boosted yield anisotropy. Frame **c** shows a heatmap of the relative anisotropy of the singlet recombination yield, $S$ (defined in Methods Eq. (10), herein quoted as a percentage), as a function of the recombination rate constants $k_S$ and $k_T$. The maximal value of $S$ amounts to 48.2%; the black dotted central diagonal indicates $k_S = k_T$. (Source Data are included as a source data file).

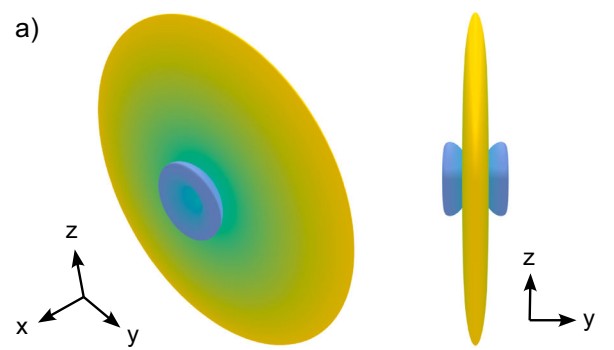

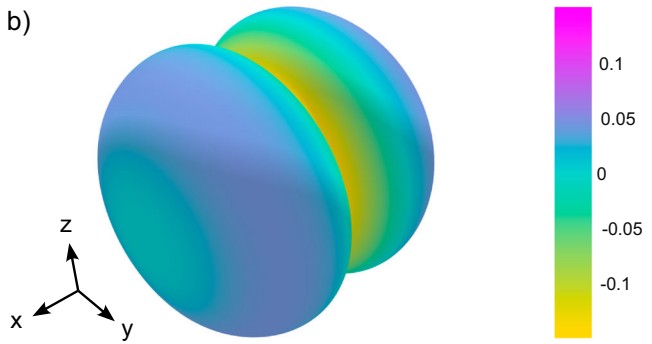

**Fig. 3 | Visualisation of the magnetic field effects (MFE) exerted on radical pairs.** Data shown here is for a toy system incorporating a single nuclear spin, the dominant hyperfine coupling with the N5 atom ($A_\perp/(2\pi) = -2.6$ MHz, $A_\parallel/(2\pi) = 49.2$ MHz), with the superoxide displaced by 4.56 Å. **a** shows the deviation of the recombination yield from its mean over all orientations, $\Phi_S(\theta, \phi) - \overline{\Phi}_S$, and **b** the singlet recombination yield $\Phi_S(\theta, \phi)$ directly. (Source Data are included as a source data file).

appear to be proportional and inversely proportional, respectively. Peak sensitivity is realised for $k_S = 3.74\,\mu s^{-1}$ and $k_T = 1 \times 10^{-3}\,\mu s^{-1}$ and amounts to a maximal change of the singlet recombination yield relative to the mean yield of $S = 48.2\%$ (or an absolute change of the recombination quantum yield of $\Delta\Phi_S = 0.246$; definitions of sensitivity measures provided in Methods). Spin relaxation from system-environment interactions has been excluded here. Figure 3 shows the directional dependence of the MFE in the geomagnetic field for auspicious rate parameters ($k_S = 3000\,\mu s^{-1}$, $k_T = 1\,\mu s^{-1}$) in terms of spherical polar plots of the reaction yield as well as the deviation of the reaction yield from the mean yield. Plots of the latter are popular in the discussion of directional MFEs[21]; the former is not usually used as the MFEs are typically too small to be discernible in this representation, unlike the effects realised here. The characteristic feature of the directional sensitivity is the reduction in the singlet recombination yield for magnetic field directions perpendicular to the dipolar axis ($x$ in this case), whereby an approximate axially symmetric pattern emerges.

Clearly, under idealised conditions and in the absence of relaxation from environmental coupling, the directional magnetosensitivity can be remarkably large despite the strong inter-radical coupling at small $r$, provided the recombination is sufficiently asymmetric.

However, it appears naïve to confine ourselves to a decoherence-free representation (aside from radical pair recombination), as even small thermally induced fluctuations of the radical positions will give rise to comparably large fluctuations of the inter-radical coupling, which are expected to induce fast spin relaxation[66]. Furthermore, superoxide is expected to exhibit a swift loss of coherence via the spin-rotational mechanism[35–37]. To assess the various potential relaxation pathways, we have extended our model to account for spin relaxation using the Nakajima-Zwanzig formalism. We have considered random field relaxation (RFR), for which the two electron spins experience uncorrelated magnetic field noise of equal amplitude in the three Cartesian directions. In addition, we have studied distance fluctuations of the two radicals, assuming that these fluctuations back-act on the spin dynamics via the distance-dependent exchange coupling or the EED interaction. We find that indeed RFR has a strongly attenuating effect on the magnetosensitivity if the coherence lifetime falls short of the required 700 ns required to realise magnetosensitivity in the weak geomagnetic field. This is seen clearly in Fig. 4, where the relative sensitivity is plotted for three values of the effective relaxation rate $\gamma = \langle\delta B_i^2\rangle\tau_C$, ranging from 0.1 to 10 $\mu s^{-1}$. Here, $\langle\delta B_i^2\rangle$ denotes the variance of the Larmor precession frequency due to the random field ($\langle\delta B_i\rangle = 0$), and $\tau_C$ is the correlation time, which was taken as 1 ns[22]. For

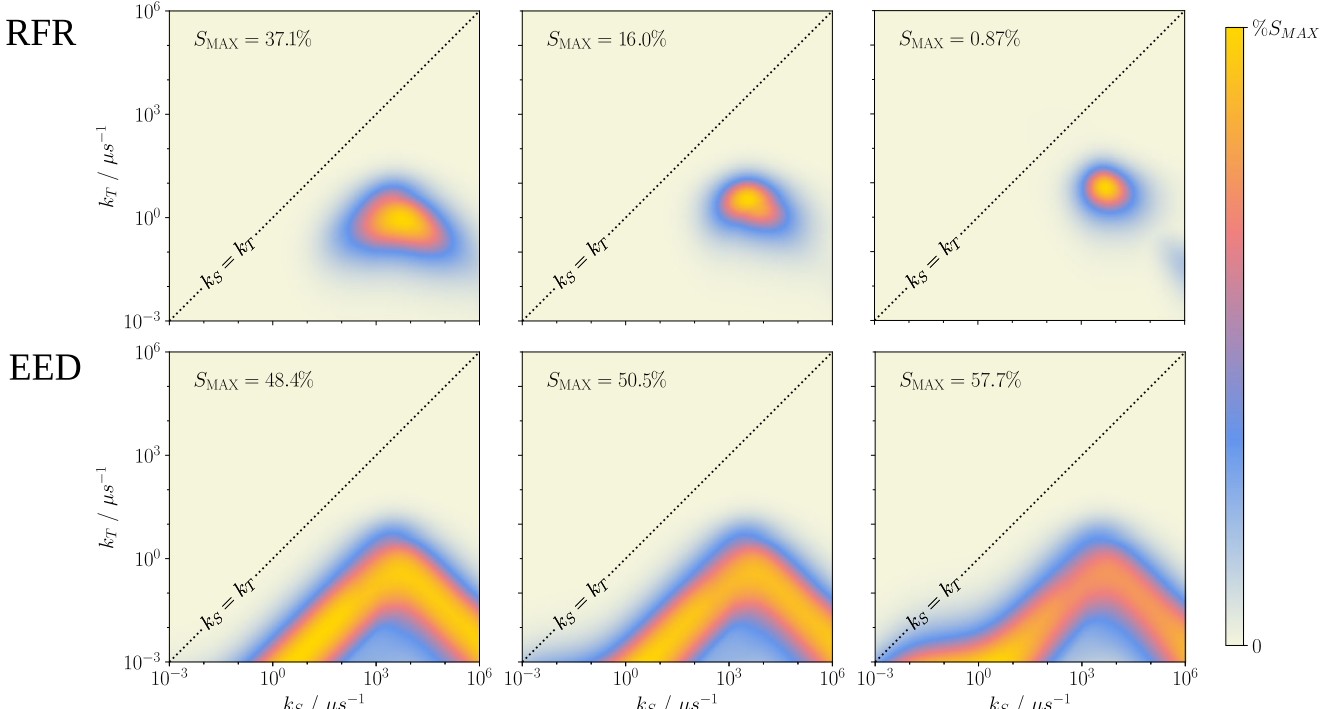

**Fig. 4 | Heatmaps of % magnetosensitivity for 2 different noise models (random field relaxation, RFR, and electron-electron dipolar coupling noise, EED) for effective relaxation rates $\gamma = 0.1\,\mu s^{-1}$, $1\,\mu s^{-1}$, and $10\,\mu s^{-1}$ (columns left to right).** The singlet yield has been sampled over 250 magnetic field orientations to evaluate the % sensitivity over a logarithmic space of singlet and triplet recombination rates $k_S$ and $k_T$. For ease of presentation, each heatmap is normalised, with the peak sensitivity value (%$S_{MAX}$) labelled on each plot. The central diagonal indicates the line of symmetric recombination $k_S = k_T$. Increasing the magnitude of EED-induced relaxation produces a comparative increase in sensitivity, up to ~58 %. However, RFR is detrimental to magnetosensitivity, at worst reducing the sensitivity to below 1 % for the largest relaxation rate. (Source Data are included as a source data file).

the largest relaxation rate, $\gamma = 10\,\mu s^{-1}$, the MFE reduces to the order of 1%, whereas for $\gamma = 1\mu s^{-1}$ MFEs of 16% are realisable for the ideally asymmetric recombination. The behaviour of the system under distance fluctuations is more encouraging. Indeed, when including EED coupling, the system appears to be resilient to distance fluctuations, actually exhibiting an enhancement of the MFE that increases with $\gamma = \mathrm{Var}(d)\tau_C$, where $\mathrm{Var}(d)$ is the variance of the dipolar coupling parameter. The largest MFE approaches 58% for $\gamma = 10\,\mu s^{-1}$, up from 48% without relaxation. Comparable enhancements of the directional MFE due to fluctuating EED interactions and enhanced singlet-triplet dephasing have been observed under symmetric recombination in well-separated radical pairs[67]. It is encouraging that comparable enhancements, or at least resilience, are observable for the strong coupling case that involves considerable fluctuation amplitude. Contrarily, fluctuations in the EED interactions appear unable to counteract the overwhelming RFR relaxation as the enhancements are realised for long radical pair lifetimes, while RFR limits sensitivity to the region of fast recombination, i.e. the tip of the triangular sensitivity pattern as a function of $\log(k_S)$ and $\log(k_T)$. Models incorporating RFR and EED-relaxation simultaneously, as well as exchange interaction, can be seen in Suppl. Fig. 6. Furthermore, we have analysed the effect of singlet-triplet dephasing (see Suppl. Fig. 7), which demonstrates that on increase of the dephasing rate $\gamma$ from 0 up to $1000\,\mu s^{-1}$, there is a negligible effect on the sensitivity enhancement, yet there is a broadening with respect to $k_S$ rates permitting the enhancement at reaction kinetics that are closer to symmetric. Similar effects are observed in the models incorporating EED-relaxation and the exchange interaction.

The results for the toy model clearly demonstrate potential for large MFEs under conditions that previously have been thought unsuitable to support such effects. We shall explore whether this is a result of merely fortuitous circumstances, realisable in symmetric and small spin systems only, or whether it could provide a viable pathway to low-field MFEs on reoxidation reactions. We shall thus investigate the effects of realistic flavin-superoxide relative positions, including their dynamics, and the impact of enlarging the spin system to include several hyperfine coupled nuclei. Based on prior insight, we expect that the latter concern is minor (hyperfine driven MFEs in FAD$^{•-}$/Z$^{•}$ are robust to the number of coupled nuclei in the flavin radical, as e.g. shown in refs. 22,34), while the former might be problematic (as the model appears to rely on a strict placement of the superoxide in the plane perpendicular to the FADH$^{•}$'s N5 principal hyperfine interaction axis, as shown above).

To assess the viability of the quantum Zeno-enhanced mechanism under more realistic conditions, we use two MD trajectories of 400 ns each[31]. Both trajectories feature a superoxide in the FAD-binding pocket, facing N5/H5 of FADH$^{•}$ and surrounded by the amino acid sidechains of Arg356, Trp395 and Asn391 (see Fig. 1 and[31] for details). According to[31], the interaction of the superoxide with the protein slows down the rotational tumbling of the molecule, whereby the rotational correlation time is increased to the order of 1 ns at 313 K (from 1 ps for superoxide tumbling freely in an aqueous solution). A closer inspection of the two trajectories revealed significant differences in placement of the superoxide, prompting us to analyse them separately. Specifically, for one of the trajectories, henceforth referred to as Trajectory 1, the superoxide was found nearly ideally placed at a position close to the FADH$^{•}$ ring plane, which we identified as potentially able to provide higher magnetosensitivity. Figure 1d shows a representation of a contour of the probability density after aligning the MD snapshots with respect to the FADH$^{•}$. The second trajectory, Trajectory 2, deviated from this ideal as the FADH$^{•}$ marginally slid backwards from the O$_2^{•-}$, tilting away from its original alignment (relative to the view

presented in Fig. 1). These transitions appear to be correlated to a reorganisation of the phosphate binding loop (PBL) of the kind described in ref. 68. Consequentially, $O_2^{\bullet-}$ assumed a position markedly below the FADH$^{\bullet}$ ring plane, facing the Arg356 residue, as is visualised in Suppl. Fig. 2 in the SI. Such a placement must be considered suboptimal for the realisation of sizeable MFEs. We note that $O_2^{\bullet-}$ is tightly enclosed by Trp395, Asn391 and Phe379, which slows down its rotational diffusion by hindering the escape of the $O_2^{\bullet-}$. An arginine (Arg356) could be particularly efficient in this regard, but the the direct front-on interaction was only seen in Trajectory 2, i.e. for the less optimal arrangements, which results following a structural rearrangement of the PBL. The current data thus hint at the structural dynamics of the PBL to be central, as it appears to regulate access to the binding pocket, enforces the optimal radical pair geometry, and gates the product release.

We investigate MFEs for both MD trajectories. To relate the observed changes in magnetosensitivity with features of the model, we have opted for a step-wise approach in which complexity (and hence closeness to the biological reality) of the system is gradually increased. Firstly, we consider the average relative FADH$^{\bullet}$/$O_2^{\bullet-}$ position and fluctuations of the relative position in three dimensions insofar as they give rise to auto-relaxation (i.e. retaining only relaxation terms that involve auto-correlations of dipolar coupling matrix elements; Model 1). Secondly, we include auto- and cross-relaxation terms with their individual covariance as derived from the MD trajectories (Model 2). These models used the point-dipole approximation to obtain the dipolar coupling tensor. In reality, the spin densities are spread out leading to deviations from the point-dipole approximation. Thus, the

third model uses an EED coupling tensor calculated by accounting for the spread of the spin densities over the FADH$^{\bullet}$ ring system and the superoxide (Model 3). For all three sets of simulations described so far, the hyperfine coupling parameters were assumed to co-align with the FADH$^{\bullet}$ molecular frame. Finally, we consider the effect of the (small) misalignment and rhombicity of the N5 hyperfine tensor principal axes system from the molecular axes system (Model 4). We use the hyperfine parameters from ref. 31, which account for thermal structure fluctuations as well as the environmental effect on the hyperfine parameters. All parameters as derived from the MD trajectories for the different scenarios are summarised in the SI section 5.

Figure 5, panels a-d summarise the MFEs found based on the Trajectory 1 parameters, presented as heatmaps showing the dependence of % sensitivity on recombination rates $k_S$ and $k_T$. When including only auto-relaxation terms based on the point dipole-dipole approximation and N5 hyperfine interaction co-aligned with the molecular frame, the peak sensitivity of 48% is realised over the range of recombination rate constants studied (Fig. 5a). This is comparable to the maximal effect sizes predicted for the toy model. Inclusion of cross-relaxation terms results in a significant reduction in sensitivity, with the maximal sensitivity falling to 8% (Fig. 5b). Using EED interaction tensors beyond the point-dipole approximation, the symmetry of the EED coupling tensor is reduced due to a non-zero $E$ parameter ($E = \frac{D_{xx} - D_{yy}}{2}$, where $D_{ii}$ are the principal values of the dipolar coupling tensor). As a consequence the sensitivity in the optimal region of Model 2 is reduced further, but the maximal sensitivity remains of the same order of magnitude at 6%, i.e. the effect on maximal effect sizes is minor (Fig. 5c). Contrary to the simpler models, this more realistic EED

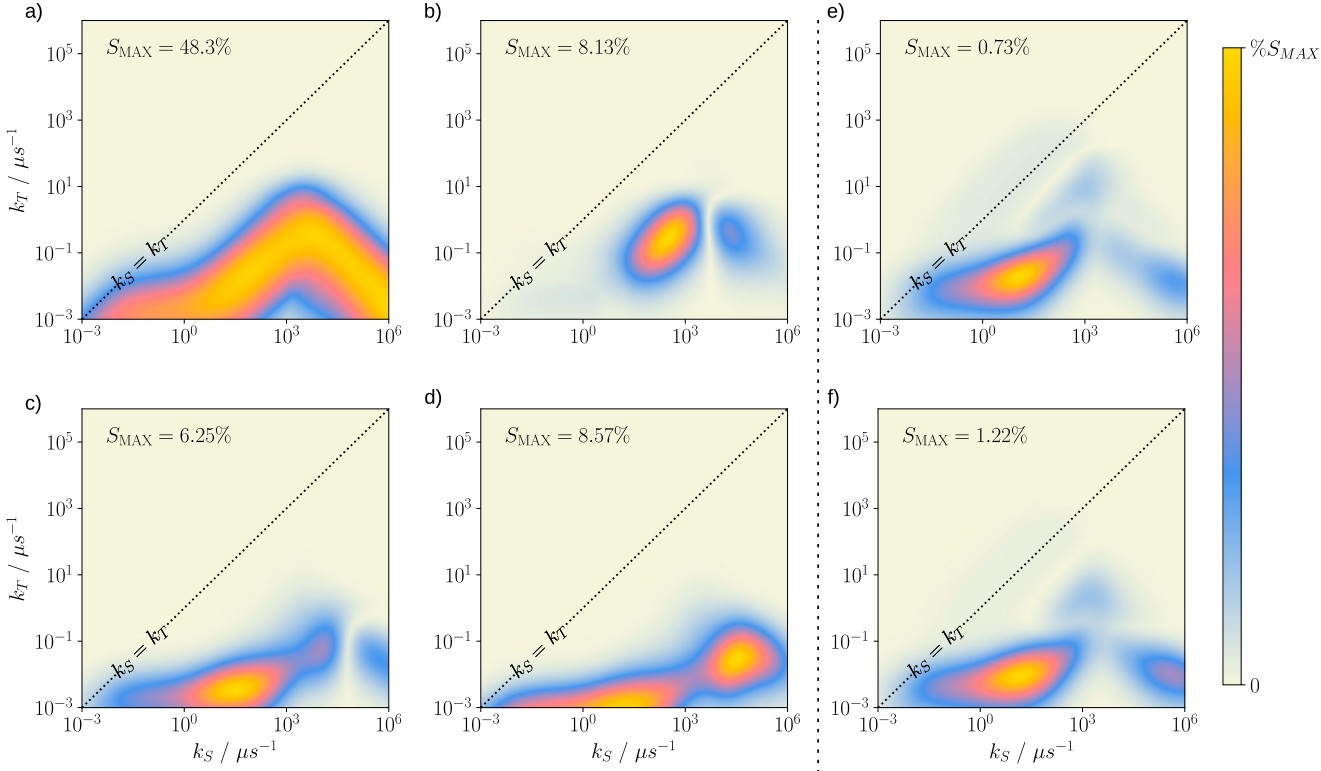

**Fig. 5 | Heatmaps of models of increasing realism and accuracy, with parameters derived from two MD trajectories for a flavin-superoxide radical pair.** Model 1 and 2 (panels **a** and **b** respectively) use the point-dipole approximation and exclude and include dipolar cross-relaxation respectively. Model 3 and 4 (panels **c** and **d** respectively) account for the delocalisation of the spin density and, in model 4 (panel **d**), the tilt of the N5-principal axes with respect to the FAD molecular axes system. Panels **e** and **f** (right of the dashed line) show results for the second trajectory: panel **e** uses model 2, and panel **f** uses model 4. These exhibit a recovery

in maximum magnetosensitivity as the system is modelled with greater complexity and biological accuracy. The plots are individually normalised to better reveal the dependence on the recombination rate constants; the peak sensitivity of each plot is indicated by %$S_{MAX}$. The central diagonal indicates the line of symmetrical radical pair recombination ($k_S = k_T$). Model 4 is the most biologically realistic, and shows a marked boost in sensitivity in the strongly asymmetrical rate ranges. (Source Data are included as a source data file).

model (Model 3) gives rise to a wider ($k_S$, $k_T$)-region of appreciable sensitivity, which covers smaller $k_T$ values and can more closely approach the symmetric recombination limit. Lastly, employing the properly aligned hyperfine interaction tensor from[69], which feature a slight misalignment relative to the molecular frame, there is a remarkable rebound in MFEs to ~8 % for low $k_T$ (Fig. 5d). In particular, deviations from the ideal geometry appear to be largely compensated by the deviation of the hyperfine interactions from their ideal. The region of peak sensitivity is located further from the central diagonal, suggesting that even more asymmetrical recombination is exploited to leverage a greater magnetosensitivity from the radical pair. Further-more, the heatmap in Fig. 5d, while predominantly exhibiting largest effects at small values of $k_T$, also indicate ample sensitivity for faster singlet recombination, for which MFE anisotropies of the order of several % are still observed (e.g. for $k_T = 2.8 \times 10^{-2} \mu s^{-1}$ and $k_S = 3.2 \times 10^4 \mu s^{-1}$, % S = 8.57%).

With regards to the second MD trajectory (with non-ideal super-oxide location), Fig. 5 panels e and f show our results for Models 2 and 4; additional results for the other models are summarized in the SI. Two key results emerge. As expected, the non-ideal superoxide loca-tion gives rise to a steep drop in maximal sensitivity to 0.73% for Model 2. Unlike for the more ideal trajectory discussed above, however, the sensitivity does not drop further with increasing complexity, but is even slightly boosted, eventually giving 1.22% for Model 4. Therefore, this shows that what was originally thought of as a strict geometric requirement, namely that $\hat{r}$ must be perpendicular to the dominant hyperfine axis, is mitigated when considering more realistic/less idealised system parameters.

To assess the robustness of this model when increasing the number of included nuclear spins, simulations for up to 5 nuclear spins were undertaken, the results of which are displayed in Suppl. Fig. 4 and Suppl. Fig. 5 in the SI. The addition of several extra spins leads to a gradual decline from 7.76% for a single spin to 0.78% for 5 spins. When using hyperfine coupling tensors transformed to a basis diagonalising the N5 tensor, 48.1% sensitivity is achieved with a single spin, but declines as more spins are included, reaching 0.58% for 5 spins. This decline slows with each additional spin, and our 5 spin model still exhibits sensitivities indicative of a functioning magnetoreceptor, suggesting an encouraging robustness of MFEs[70].

## Discussion

The FADH•/O$_2^{•-}$ model enjoys widespread popularity, and has been suggested to account for various magnetosensitive traits in biology, including avian magnetoreception and neurogenesis[43,44,71]. Whilst it could thus revolutionise our understanding of magnetic field effects in biology, potentially relevant to physiology, these claims are not with-out controversy, in particular, when they involve weak magnetic fields such as the geomagnetic field, and are not presently supported by direct experimental evidence. The only direct evidence of such MFEs pertains to RPs involving (unbound) superoxide in exceedingly high strength magnetic fields, i.e. fields vastly exceeding the geomagnetic field[72]. One root of the controversy around FADH•/O$_2^{•-}$ is the fast spin relaxation of superoxide, arising as a result of the large spin orbit interaction through the spin rotational mechanism when the anion tumbles freely in solution. Such fast spin relaxation nullifies the spin correlation of the radical pair, rendering the recombination reaction insensitive to weak magnetic fields. As the relaxation rate is directly proportional to the anion's rotational correlation time, it is often thought that an easy fix could be found through its immobilisation[54]. While this might indeed be the case as far as spin relaxation is con-cerned, the argument overlooks the fact that immobilisation at a dis-tance that permits spin-sensitive recombination reactions necessarily implies inter-radical distances at which the EED interaction is domi-nant. In such a case, prior models suggested that the coherent singlet-triplet interconversion would be blocked, again leading to no marked

weak-field magnetosensitivity. This dilemma was occasionally recog-nised, and several models have been suggested to counteract the EED interaction by the Hore group and us. However, these measures have been either found unproductive for FADH•/O$_2^{•-}$ or rely on three-radical effects, thus implying a high level of complexity to orchestrate the properties and availability of the third radical[24,38]. The model sug-gested here, relying on regaining sensitivity through inducing the quantum Zeno effect with asymmetrical recombination rates, is thus attractive as it is engineered around a simple radical pair recombination.

The magnetic field sensitivity observed here for FADH•/O$_2^{•-}$-type radical pairs immobilised at what is essentially the contact distance is remarkable. However, the model is not without its foibles. The largest effects were predicted for simple, symmetric one-nitrogen systems with an idealised axial hyperfine interaction perpendicular to the EED interaction axis. Even though the hyperfine interactions in FADH• are dominated by the N5 nucleus, a marked drop in the predicted sensi-tivity was evident when increasing the number of coupled nuclear spins in FADH•. This is tempered by the presence of appreciable MFEs for a 5-spin system of near one percent for the symmetric model. While such effects sizes might appear small relative to the 58% predicted under idealised conditions, one must not forget that directional MFEs of realistic biological radical pairs in the geomagnetic field are gen-erally predicted to be small. For example, the better accepted FAD/W model predicts MFEs on the order of only 0.01% for realistically com-plex radical pairs, even if EED interactions are precluded from the calculation[19,73,74]. A further reduction by as much as an order of mag-nitude is expected if EED interactions were included[24].

Cryptochrome could be a potential host of a magnetosensitive FADH•/O$_2^{•-}$ radical pair, thus facilitating the avian magnetic compass and more broadly cryptochrome based magnetic field effects[43]. The FAD is non-covalently bound to the protein by intercalating it betwixt two alpha helices and partly shielding it by flexible loops, the phosphate-binding loop and interface loop, to establish a snugly enclosed binding pocket. Superoxide contained in this binding pocket at a location identified in previous MD studies facing the N5/H5 of FADH• appears to be well placed to induce large MFEs by the described mechanism. Furthermore, fluctuations of the EED interaction due to the anion's thermal motion in the binding pocket were found to not inhibit sensitivity. Where the placement of the superoxide deviated from the idealised, symmetrical model (e.g. as a consequence of pro-tein rearrangements as seen for Trajectory 2), the model retained ample sensitivity when employing more realistic hyperfine coupling parameters and extending beyond the point-dipole approximation for the EED interaction.

We note that the fold and residues surrounding the FAD in its putative magnetosensitive configuration are highly conserved across the cryptochrome family, suggesting that familiar results would be expected in cryptochromes that bind FAD and can be fully reduced. Specifically, avian Cry1 and Cry4 are not expected to show funda-mental differences from the point of view of this mechanism. In the SI, we compare different cryptochromes with respect to the structure of homology, including all cryptochromes for which dark state magnetic field effects have been described[16,75,76] (see Suppl. Fig. 13 & Suppl. Fig. 14).

While some aspects of the model presented here are auspicious, it would be too early to unequivocally accept that FADH•/O$_2^{•-}$ can indeed be markedly magnetosensitive as a result of radical pair recombination in weak magnetic fields. Several aspects of the overall processes require more precise definition and assessment before reaching a conclusion on the potential magnetosensitivity. One of these pertains to spin relaxation of superoxide to which the mechanism is not robust, i.e. the random field relaxation as it effectively manifests as a result of the spin rotational interaction mechanism and other relaxation path-ways. Here, a recent MD study suggests that the rotational dynamics of

superoxide in the binding pocket are sufficiently arrested so as to permit MFEs[31]. However, it is still unclear if and how this constraint could be realised throughout the entire reaction processes whilst still permitting radical escape or another alternative pathway besides singlet state recombination, as required to realise magnetic field sensitivity. While superoxide appears to be reasonably well placed and rotationally impeded to allow MFEs in principle, the available MD data do not exhibit indications of competing radical escape processes. In this regard it is interesting to note that a recent study has identified a gate-like function of the phosphate binding loop, which appears to allow access to the FAD binding pocket in a FAD redox-state dependent fashion[31]. One could thus envisage a magnetosensitive radical pair that is snugly bound in the binding pocket recombining in a magnetic field-dependent manner through the delineated mechanism until the 'gate' opens. Upon which event, superoxide escapes into the bulk while the protein undergoes a rearrangement to signal the reaction outcome to subsequent processes. Another issue related to relaxation is the preference of the more realistic FADH·/O$_2^{·-}$ radical pairs to favour very strongly asymmetric recombination, as evidenced in the studied MD trajectories. This inevitably leads to long-lived radical pairs which in turn become correspondingly more susceptible to spin relaxation. This suggests that a compromise will have to be struck between large intrinsic effects and the radical pair lifetime. Consequently, only a fraction of the predicted sensitivity might become usable under realistic conditions. Overall, whilst we demonstrate the promising possibility of significant quantum Zeno enabled MFEs, calculations for more accurate models will be necessary to better deduce the actual limits of sensitivity as a result of these caveats, but these are beyond this first exploration of principle feasibility.

Finally, we ask what is at the root of the remarkable effect discussed here? In section 3 of the SI, we discuss a one-nucleus toy model that captures essential traits, while permitting analytical insights. To this end, we assume that the hyperfine interaction obeys $A_{xx}=A_{yy}=A_\perp=0$ and $A_{zz}=A_\parallel \gg \omega$, which is approximately fulfilled for N5 in FADH, and that the EED axis is parallel to the $x$-axis. The dynamics of this system in the absence of relaxation can be described in terms of a reducible effective, non-Hermitian 'Hamiltonian' that accounts for the combined effects of coherent evolution and singlet recombination. The slow triplet recombination can be disregarded at first (this is formalised in the SI). Its irreducible blocks can be analysed using non-Hermitian perturbation theory. As demonstrated in the SI, the effects described here result from the quantum Zeno effect, as identified and

defined generally[51,53,77,78] implying the effect from asymmetric recombination without requiring a strict quantum measurement interpretation[49] (i.e. including environment interactions, dissipative processes[79] and generalised quantum operations[51]) and present in all common treatments of the recombination. This is further supported by our results, where a $1/k_S$ scaling of the imaginary parts of the effective Hamiltonian emerges, e.g. visible in the density plot in Fig. 2, that is indicative of the quantum Zeno effect. The remarkable magnetosensitivity then results, for triplet-born radical pairs, from a fortuitous degeneracy of states, induced by the quantum Zeno effect, that are or are not coupled by the Zeeman interaction to the recombining manifold depending on the direction of the magnetic field. We note that this is not a mere effect of lifetime broadening. The lifetime of a state receptive to the geomagnetic field must be on the order of $1\,\mu s$ (such that $\omega\tau \simeq 1$, where $\omega$ is the Larmor precession frequency in the geomagnetic field), regardless of the recombination rate constants. However, the lifetime broadening of such a state is only on the order of 1.6 MHz, which is tiny relative to the energy level differences resulting from the electron-electron dipolar interaction (~1 GHz) in a tightly bound pair.

The large magnetosensitivity observed here then originates from a stark contrast in recombination yield when the field is parallel to $x$ compared to $y$ or $z$. For the magnetic field along the $z$-direction, the $|S\rangle$ and $|T_0\rangle$ states are uncoupled from the remaining triplet states. The latter will thus not contribute to the singlet recombination yield. The dynamics of the $|S\rangle$, $|T_0\rangle$ block are governed by the quantum Zeno effect. For the magnetic field along the $x$-direction, on the other hand, all states are coupled. The magnetic field connects the $|T_0\rangle$ and $|T_\Sigma\rangle$ basis states, which are degenerate eigenstates of the dominant EED interaction. Thus, under the quantum Zeno effect, an efficient coherent interconversion becomes possible between the $|T_0\rangle$, $|S\rangle$-manifold and the $|T_\Sigma\rangle$ state. Different from the situation with the magnetic field along the $z$-direction, we here end up with two, instead of one, states that depopulate at a rate determined by the quantum Zeno effect, besides the S-like state that is quickly depopulated in both cases. It is this contrast that ultimately gives rise to the large directional MFEs observed for this system. Figure 6 illustrates the dependence on the imaginary part of the eigenvalues as derived from the perturbation expressions as a function of $\kappa$, i.e. after removal of the triplet recombination. It is apparent that the resulting dependence directly mirrors the triangular pattern seen in the MFEs. This is so because maximal magnetosensitivity results if the triplet recombination rate is

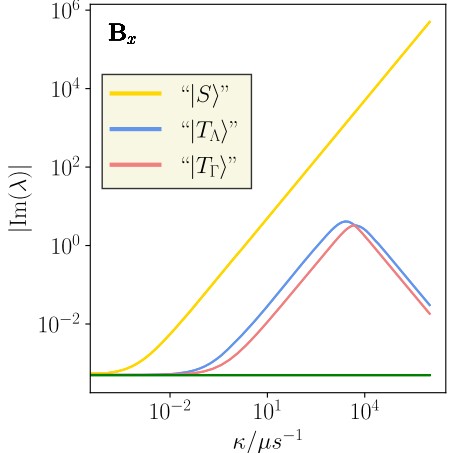
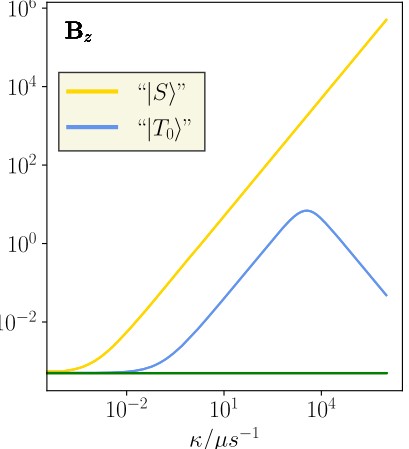
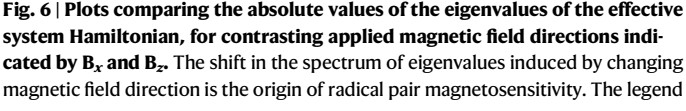

**Fig. 6 | Plots comparing the absolute values of the eigenvalues of the effective system Hamiltonian, for contrasting applied magnetic field directions indicated by B$_x$ and B$_z$.** The shift in the spectrum of eigenvalues induced by changing magnetic field direction is the origin of radical pair magnetosensitivity. The legend indicates what states are signified by the different coloured lines. Inverted commas are used in order to indicate the 0$^{th}$ order state the line is associated with, but that each line does include small contributions from other states. $|T_\Lambda\rangle = \frac{1}{\sqrt{2}}(|T_\Sigma\rangle + |T_0\rangle)$ and $|T_\Gamma\rangle = \frac{1}{\sqrt{2}}(|T_\Sigma\rangle - |T_0\rangle)$. (Source Data are included as a source data file).

comparable to the effective depopulation rates as contained in the eigenvalues of the effective Hamiltonian, as a balanced scenario of effective singlet and triplet recombination maximises the impact of changes in the coherent evolution/reaction dynamics.

We have herein identified a scenario for which RPs at close contact may in principle develop remarkable magnetic field sensitivity in weak magnetic fields when immobilised in rigid or semi-rigid matrices, such as being bound in a protein. This is despite being subjected to strong inter-radical coupling, such as the unavoidable electron-electron dipolar interaction. The phenomenon relies on the EED interaction as the dominant interaction to impose an ideal energy level structure that responds to comparably tiny magnetic fields when within the quantum Zeno regime (as realised by fast asymmetric recombination). This can serve as a blueprint for realising large directional MFEs in weak magnetic fields. Our employment of Nakajima-Zwanzig equations of motion to investigate the 'Zeno regime' of the RPM proved to be accurate and computationally efficient. The model is characterised by simplicity and relies on requirements that appear to be readily engineerable and controllable in synthetic or biomimetic systems, such as donor-acceptor dyad molecules, and arguably commonplace in biological systems[80]. Specifically, key ingredients are the triplet initial state, fast recombination via the singlet channel, and a dominant dipolar interaction perpendicular to the main hyperfine quantisation axis. Randomly encountering radical pairs (F-pairs) will show comparable responses following the fast recombination of the initial singlet population. For randomly oriented samples, e.g. frozen solutions or immobilised randomly oriented samples, the effect will support a magnetic field intensity dependence of the recombination yield in weak magnetic fields. Bound flavins, such as FAD cofactors non-covalently bound to redox enzymes, are viable candidates to exhibit magnetic field effects following this mechanism. The reoxidation of fully reduced flavins with molecular oxygen, yielding FADH$^\bullet$/O$_2^{\bullet-}$, could acquire magnetosensitivity if the superoxide is sufficiently immobilised to attenuate spin rotational relaxation pathways. Unlike the RPM in the usual regime of balanced recombination ($k_S \sim k_T$), which is generally assumed to yield maximal spin-state (and thus magnetic field) sensitivity, the proposed mechanism is not suppressed by the overwhelming EED interactions that are necessarily present for radical pairs in close contact. On the contrary, the EED interaction is a key constituting element of the mechanism. Furthermore, the decoherence pathways induced by modulations of the EED and exchange interaction resulting, for example, from residual radical motion in a binding pocket are not necessarily destructive to the MFEs, and can even be beneficial in some cases.

While the mechanism developed here suggests an exciting opportunity for MFEs to be realised in biological processes, the question of whether FADH$^\bullet$/O$_2^{\bullet-}$ in cryptochromes is a viable magnetoreceptor remains open. In principle, the suggested mechanism can deliver the required magnetosensitivity. Yet stringent requirements must be met, in particular concerning the degree of immobilisation of O$_2^{\bullet-}$ possible whilst permitting alternative reaction pathways other than recombination (such as superoxide escaping into the bulk). While the answers to these far-reaching questions will require more concrete and detailed models, ideally motivated from direct experimental observations, the mechanism as suggested solves the previously perceived dilemma of the currently established RPM of requiring rigid immobilisation whilst simultaneously having to avoid strong inter-radical coupling—an impossible balancing act. We hope that the proposed model will lead to a surge in the research of magnetic field effects in reoxidation reactions and provide more clarity about magnetic field effects in biology. We want to emphasise that, at the current stage, this work is not presenting superoxide-based RPs as the pre-eminent facilitators of magnetoreception, but rather highlights the potential for considerable directional magnetosensitivity in strongly coupled systems, enabled by the quantum Zeno effect.

## Methods

We focus on the spin density operator $\hat{\rho}(t)$ of the radical pair. Its dynamics, subject to coherent evolution, radical pair recombination and spin relaxation in the Markovian limit, are modelled by the master equation of the type[60]

$$\frac{\mathrm{d}}{\mathrm{d}t}\hat{\rho}(t) = \hat{\hat{\mathcal{L}}}\hat{\rho}(t) + \hat{\hat{\mathcal{R}}}_{NZ}\hat{\rho}(t)$$
$$= -\mathrm{i}\left[\hat{H}, \hat{\rho}(t)\right] - \left\{\hat{K}, \hat{\rho}(t)\right\} + \hat{\hat{\mathcal{R}}}_{NZ}\hat{\rho}(t), \quad (1)$$

where $[\cdot,\cdot]$ and $\{\cdot,\cdot\}$ signify the commutator and an anti-commutator, respectively, and we have subsumed the effects of coherent evolution and recombination in the Liouvillian $\hat{\hat{\mathcal{L}}}$ for later reference.

The time-averaged spin Hamiltonian $\hat{H}(\hbar = 1)$ governs the radical pair's coherent evolution, and includes the following terms consists of the following terms:

$$\hat{H}_{\mathrm{HFC}} = \sum_{k=1}^{N_1} \hat{\mathbf{I}}_{1,k} \cdot \mathbf{A}_{1,k} \cdot \hat{\mathbf{S}}_1 \quad (2\mathrm{a})$$

$$\hat{H}_{\mathrm{Dipolar}} = -\sum^{m} d_{1,2}(|\mathbf{r}_{1,2}|)(3(\hat{\mathbf{S}}_1 \cdot \mathbf{1}_{1,2})(\hat{\mathbf{S}}_2 \cdot \mathbf{1}_{1,2}) - \hat{\mathbf{S}}_1 \cdot \hat{\mathbf{S}}_2), \quad (2\mathrm{b})$$

$$\hat{H}_{\mathrm{Zeeman}} = -\gamma_e \mu_B \sum_{i=1,2} \hat{\mathbf{S}}_i \cdot \mathbf{B}, \quad (2\mathrm{c})$$

where $\hat{\mathbf{S}}$ and $\hat{\mathbf{I}}$ denote spin operators for electron and nuclear spins respectively, $\mathbf{A}$ is a hyperfine coupling tensor, and $\mathbf{1}_{i,j}$ are elements of a unit vector along the dipolar axis. Constants take their usual definitions, and $d_{i,j}(r) \equiv \mu_0 g_e^2 \mu_B^2 / 4\pi r^3 > 0$ with $r \equiv |\mathbf{r}_{i,j}|$. As we consider a FADH$^\bullet$/O$_2^{\bullet-}$ model, where only the FADH$^\bullet$ has non-zero hyperfine couplings, $\hat{H}_{\mathrm{HFC}}$ includes no couplings relating to O$_2^{\bullet-}$.

The second term on the second line of equation (1) accounts for spin selective recombination in the Haberkorn framework[63]. Assuming that the radical pair reacts with first-order rate constants $k_S$ and $k_T$ in the singlet and triplet states, respectively, the associated Haberkorn reaction operator $\hat{K}$ is given by

$$\hat{K} = \frac{k_S}{2}\hat{P}_S + \frac{k_T}{2}\hat{P}_T, \quad (3)$$

where $\hat{P}_S$ and $\hat{P}_T$ denote the projection operators on the singlet and triplet states, respectively. In order to incorporate relaxation processes into the study of the quantum Zeno scenario, we follow the suggestion of Fay et al. and utilise an approach derived from the Nakajima-Zwanzig master equation in the Schrödinger picture (unlike the Redfield relaxation tensor, which results from an analog approach in the interaction representation), which is concisely contained in the relaxation superoperator $\mathcal{R}_{NZ}$[60–62]. Although also of second order in the system-bath interaction, the Nakajima-Zwanzig approach has proven far more accurate than standard Redfield-theory when applied to radical pairs undergoing asymmetrical recombination − indicative of the quantum Zeno regime − and appears to be applicable for stronger system-bath coupling.

For a system-bath interaction Hamiltonian of the form $H_1 = \sum_i X_i(t)\hat{A}_i$, where $X_i$ is a random variable reflecting the protein's thermal motion and its coupling to the spin system ($\langle X_i \rangle = 0$), and $\hat{A}_i$ an operator in the system Hilbert space, $\hat{\hat{\mathcal{R}}}_{NZ}$ can be expressed as

$$\hat{\hat{\mathcal{R}}}_{NZ} = -\sum_{j,k} \int_0^\infty d\tau\, g_{j,k}(\tau) \hat{\hat{A}}_j^\dagger e^{\hat{\hat{\mathcal{L}}}\tau} \hat{\hat{A}}_k, \quad (4)$$

where $g_{j,k}(t)$ is the time correlation function of $X_j$ and $X_k$ such that

$$g_{j,k}(t) = \left\langle X_j^*(0) X_k(t) \right\rangle \tag{5}$$

and $\hat{\bar{A}}_i$ denotes the commutation superoperator associated with $\hat{A}_i$, i.e. $\hat{\bar{A}}_i \cdot = [\hat{A}_i, \cdot]$.

Given that $\hat{\mathcal{L}}$ can be expressed as $\hat{\mathcal{L}} = -i\left(\hat{H}_{\text{eff}} \cdot - \cdot \hat{H}_{\text{eff}}^\dagger\right)$ with $\hat{H}_{\text{eff}} = \hat{H} - i\hat{K}$, the equation of motion (Eq. (1)) can be conveniently evaluated in the non-unitary eigenbasis of $\hat{H}_{\text{eff}}$, for which Eq. (4) assumes the form of a sum over products of matrix elements of $\hat{\bar{A}}_i$ and $\hat{A}_j$ multiplied by spectral densities

$$j(\omega) = \int_0^\infty dt\, g(t) e^{i\omega t} \tag{6}$$

evaluated at the eigenvalue differences of $\hat{H}_{\text{eff}}$[81]. For random variables with exponential correlation functions, $g_{j,k}(t) = \langle X_j^* X_k \rangle \exp(-t/\tau_c)$, as assumed here for simplicity, $j(\omega)$ assumes the form $j_{j,k}(\omega) = \langle X_j^* X_k \rangle (\tau_c^{-1} - i\omega)^{-1}$. Here, $\tau_C$ is the correlation time of the system, and $\langle X_j^* X_k \rangle$ is the covariance (mean squared fluctuation for $j = k$) of the system-environment couplings.

$\hat{H}_1$ could take various forms depending on the system-bath interaction under consideration. For example, The random field relaxation Hamiltonian

$$\hat{H}_{\text{RFR}}(t) = \sum_{i=x,y,z} \sum_{j=1,2} \mathbf{b}_{i,j}(t) \cdot \hat{\mathbf{S}}_{i,j}, \tag{7}$$

models stochastic fluctuations in all axes ($i \in \{x, y, z\}$) for each electron in the radical pair ($j \in \{1, 2\}$). Whilst the Hamiltonian modelling the exchange interaction between the electrons is as follows

$$\hat{H}_{\text{EXR}}(t) = -2J(r)\hat{\mathbf{S}}_1 \cdot \hat{\mathbf{S}}_2, \tag{8}$$

where $\hat{\mathbf{S}}_1$ and $\hat{\mathbf{S}}_2$ represent the spins of the two radicals, and $J(r)$ is the exchange coupling strength dependent on $r(t)$.

In the SI, we evaluate the accuracy of the approach for several relevant scenarios by comparing against results calculated with the hierarchical equation of motion (HEOM)[64,82], which is numerically exact. For the relaxation processes and parameter ranges relevant to this study the Nakajima-Zwanzig approach is found suitable without reservation, even for strongly asymmetric recombination and large fluctuating EED coupling at the contact distance (see Suppl. Fig. 16 and Suppl. Fig. 17 in the SI). Here, we consider spin relaxation due to uncorrelated random field fluctuations, i.e. random field relaxation (RFR) − a widely used generic approximation of various inter and intramolecular spin relaxation mechanisms, including spin rotation, and due to the fluctuation of inter-radical interactions as arising from the radicals' thermal motion in the binding pocket[83]. As a result of comparably small inter-radical distances and correspondingly large interaction strengths, we expected the latter to be particularly relevant, much more so than for well separated radical pairs, such as FAD$^{\bullet-}$/W$^{\bullet+}$. To obtain realistic estimates of the electron-electron dipolar couplings and their relaxation-inducing fluctuations, we extracted the relevant coupling parameters and their covariances from molecular dynamics simulations. The molecular dynamics trajectories, which were taken from a previous work[31], used the cryptochrome 4a structure[84], as available from the Protein Data Bank under accession code 6PU0. The initial and final structures are also available from the Supplementary Data 1.

In order to assess the directional magnetosensitivity of the radical pair systems in the geomagnetic field, we evaluate the singlet recombination yields from

$$\Phi_S = k_S \int_0^\infty dt\, \text{Tr}\left\{\hat{P}_S \hat{\rho}(t)\right\}, \tag{9}$$

for 300 magnetic field field orientations to find the sensitivity:

$$S = \frac{\Delta \Phi_S}{\overline{\Phi}_S} = \frac{\Phi_{S,\text{MAX}} - \Phi_{S,\text{MIN}}}{\overline{\Phi}_S}, \tag{10}$$

where $\overline{\Phi}_S$ denotes the mean of the recombination yield.

Code to calculate the singlet recombination yield was implemented in Python using NumPy[85], SciPy[86], and QuTip[87,88], the latter to generate the spin operators. Yields were calculated by evaluating the time-integrated density operator $\int_0^\infty \rho(t)dt$ by solving the linear system corresponding to the Laplace transform of eq. (1) for Laplace parameter $s = 0$. Simulation parameters are provided in the Supplementary Information.

## Reporting summary

Further information on research design is available in the Nature Portfolio Reporting Summary linked to this article.

## Data availability

The data generated in this study are provided in the manuscript, the Supplementary Information, Supplementary Data, and Source Data files. Source data are provided with this paper.

## Code availability

The code underpinning this publication has been made available via Code Ocean. The associated Compute Capsule's DOI is https://doi.org/10.24433/CO.3355578.v1.

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

## Acknowledgements

This work was supported by the Office of Naval Research Global (ONR-G Award Number N62909-21-1-2018), the EPSRC (EP/X027376/1 and EP/V047175/1), and BBSRC (BB/Y51312X/1). We acknowledge use of University of Exeter's high-performance computing facility. For the purpose of open access, the author has applied a Creative Commons Attribution (CC BY) licence to any Author Accepted Manuscript version arising from this submission.

## Author contributions

D.R.K. conceived the study. All authors contributed to the research through conducting computer simulation, data analysis, or analytic derivations. D.R.K., L.D.S., and M.C.J.D. developed the underpinning software. J.P. and A.T. contributed as part of their Master's thesis. D.R.K. and L.D.S. supervised the work. W.X. contributed as part of an Erasmus+ placement, particularly on the perturbation expansions of the effective Hamiltonian. D.R.K., L.D.S., and M.C.J.D. wrote the manuscript, with inputs from co-authors. D.R.K. acquired the financial support for the project, and managed and coordinated the research activities.

## Competing interests

The authors declare no competing interests.
