## [Transparent Peer Review file · Nature Communications]

Magneto-sensitivity of Tightly Bound Radical Pairs in Cryptochrome is Enabled by the Quantum Zeno Effect

Corresponding Author: Professor Daniel Kattinig

Version 0:

Reviewer comments:

Reviewer #1

(Remarks to the Author)

The authors study in depth the repercussions of the Quantum Zeno effect in the radical-pair mechanism. Specifically, they invoke asymmetric recombination rates in a realistic radical-pair system, namely FAD-superoxide. They find, counterintuitively, that when the radicals are in close proximity, with the dipole interactions and the recombination rates being significantly enhanced, the particular choice of asymmetric recombination rates, for which choice the Quantum Zeno effect is readily manifested, leads to unexpectedly high magneto sensitivity.

Although quantum measurement dynamics, and concomitantly the Quantum Zeno effect in the radical-pair mechanism have been previously discovered and analyzed (Refs 45 and 46), this study goes in far-reaching depth using a realistic radical-pair model, and taking advantage of the Quantum Zeno effect in a much broader biophysical setting involving a full description of dipole interactions, spin-orbit interactions and the resulting spin relaxation.

This study opens up an amazing new window into the possibilities offered by the so-called "non-trivial" quantum effects in biology. The repercussions of this study could well reach beyond the specific problem of magneto sensitivity. The manuscript is extremely well written. Overall, I happily recommend publication.

There are few issues the authors need to address:

1) The authors might want to shortly comment on why asymmetric recombination rates are realistic for this particular radical-pair model, and also comment, perhaps in the context of previous work on electron transfer reactions with this system, about the particular numerical values of the recombination rates deep in the Quantum Zeno regime.

2) Along these lines, since the authors consider a triplet born radical-pair, they could cite the paper "Coherent Triplet Excitation Suppresses the Heading Error of the Avian Compass" at New Journal of Physics 12, 085016 (2010), where it is discussed that triplet born radical-pairs seem to be advantageous. The authors could then shortly estimate what fraction of the magneto-sensitivity they calculate is due to the triplet initial condition, e.g. by repeating the calculation for the opposite values of the recombination rates and starting from the singlet state.

3) In their toy model elaboration on the Quantum Zeno effect, it would be helpful, especially for the general reader, to produce some plots with time in the x-axis, in order to elucidate spin dynamics and connect with the rest of the discussion.

4) Since the authors use the Haberkorn recombination operators, it would further be helpful if the authors elaborate on what is the effect of the zero-quantum coherence relaxation term $-(k_S+k_T)(Q_S.p+p.Q_S-2Q_S.p.Q_S)$ discussed in Ref 45 and follow up references. Is this term hidden in Nakajima Zwanzig formalism? If not, would this term alter the main results on spin-relaxation and magneto sensitivity? I do not propose to the authors to get involved in master equation discussions, but S-T relaxation is significant in this strong Quantum Zeno regime and it would be very helpful if it is clearly understood, or if the specific assumptions made in the authors' calculations are clearly defined.

5) The authors mention previous work of this reviewer (Refs 45 and 46), but Ref 45 does not deal with magneto-sensitivity loss and its recovery in the Quantum Zeno regime. Ref 45 introduces quantum measurement dynamics in the radical-pair mechanism, and the resulting Quantum Zeno effect in the regime of asymmetric recombination rates. This ought to be clearly

stated, i.e. citations 45 and 46 ought to be decoupled.

6) In the caption of Figure 5 there is in the ket T_{λ} , but in the left figure there is T_{Δ}

Iannis Kominis

(Remarks on code availability)

Reviewer #2

(Remarks to the Author)

One of the major unsolved questions in the field of biological magnetoreception is the mechanism whereby chemical magnetosensing occurs in biological systems, as distinct from ferromagnetite based polarity compass mechanisms. This so-called Radical Pair mechanism involves spin chemical forces which alter reaction rates of biological signaling pathways and thereby result in a directional response to applied magnetic fields. Much physiological and behavioural evidence has implicated flavoproteins, known as cryptochromes, as the likely receptors in this process in the avian system as well as several others. Although it is further demonstrated that biological magnetosensing requires illumination of cryptochrome, the actual magnetosensitive reaction step occurs in the dark, during the process of flavin reoxidation, concomitantly with the formation of the $FADH^{\bullet}/O^{\bullet-}$ 2 RP, which has therefore been an obvious candidate for the magnetosensitive radical pair. However, the deleterious effects of spin relaxation arising through the spin rotational mechanisms, and of electron-electron dipolar (EED) coupling of the radical pairs at short inter-radical distances on the mag-3 netosensitive dynamics, severely limit the theoretical response of this system to near-earth strength magnetic fields of the type that birds and biological organisms respond to. Therefore, both the nature of the radical pair and of how they respond in the magnetic field has remained a mystery.

Here the authors report for the first time a possible theoretical solution to this continuing conundrum by investigating the influence that the so-called quantum Zeno effect could exert on a triplet initialised flavin semiquinone-superoxide system such as found in cryptochromes. RPs at close contact may in principle develop remarkable magnetic field sensitivity in weak magnetic fields when immobilised in rigid or semi-rigid matrices, such as being bound in a protein despite being subjected to strong inter-radical coupling, such as the unavoidable electron-electron dipolar interaction. This mechanism solves the dilemma of currently established radical pair mechanisms that require rigid immobilisation while simultaneously having to avoid strong inter-radical coupling – a physical impossibility. Therefore this mechanism not only introduces a new paradigm for exploration of the Radical Pair mechanism in general, but also for the first time provides a theoretical explanation for magnetosensing that actually fits all of the known experimental data for magnetic orientation in the avian model, as well as in others that occur independently of light. Given that this model can also be applied to other proposed magnetosensing systems including in the mitochondria and in the human brain, this theoretical approach can be of wide spread interest to several different disciplines and, even more importantly, provides a clear blueprint for experimental verification. As lack of a testable, physically sound theoretical mechanism for magnetosensing at near-earth strength has been the single greatest factor holding up progress in the field of biological magnetosensing, I welcome this submission as an important aid to further progress and insight.

I have just a few minor comments and corrections to bring things more in line with biological data. These authors use the pigeon Cry4 in their experimental model and implicitly identify this as an avian magnetosensor. However the Cry4 cryptochrome undergoes a very slow flavin redox cycle (half-life for flavin reoxidation is 2 - 6 hours, depending on species) similar to photolyases and distinct from sensory Crys such as plant and fly Crys. This is incompatible with a signaling (orientation) role as birds orient rapidly (pigeons take off in the homing direction within 5 minutes). Furthermore the avian Cry4 is found in retina in a cell type that contains oil droplets, which renders them opaque to UV light. Since birds orient best in UV, this is incompatible with avian navigation (see eg. Wiltschko et al, *Frontiers in Physiology* 2021; some discussion also in Aguida et al, *Frontiers in Plant Physiology* 2024). In the absence of any supporting experimental evidence for Cry4 as a magnetosensor the authors should state explicitly in the paper that Cry4 was used in their modelling (for convenience?) but is incompatible with current behavioural and histological evidence as having a role in avian magnetosensing. The paper would also be improved and have broader appeal if the authors could discuss their results more explicitly with respect to biological effects and known magnetosensing cryptochromes (eg drosophila and plant crys, which are proven to have magnetosensing function). For example discuss what kind of changes in reaction rate one might expect at different field strengths (rate of flavin reoxidation? Low Level Field? should rates increase or decrease?) with this model. Also, what kinds of other physiological parameters or structural changes in the protein might be affecting these rates (dimerisation? cellular redox potential?) and what kinds of residues/structural features/ amino acid oxygen binding sites might be particularly important for these mechanisms? One intriguing question is whether a scavenging mechanism could occur in combination with the Zeno effect to further amplify the MFE.

(Remarks on code availability)

Reviewer #3

(Remarks to the Author)

This paper shows that asymmetry effects in spin-selective chemical reactions play a major role in their magnetic field effects and their anisotropy. To make this more concrete, calculations are performed using a model of the reaction between cryptochrome and superoxide, a candidate molecule for magnetosensitivity in migratory birds.

Although the possibility of a cryptochrome-superoxide model has been demonstrated by some of the authors of this paper, the aspect of the binding between oxygen and cryptochrome has not been experimentally demonstrated except by molecular dynamics simulations, and the formation of radical pairs is completely unknown.

Although the reviewer has some doubts about the necessity of detailed calculations for this unknown radical pairing, the reviewer acknowledges that this is an important model calculation in terms of the effect of spin-selective chemical reactions on magnetic field effects.

The model calculations presented here are very precise, and from the reviewer's point of view, the calculations are highly sophisticated, and the interpretation of the model calculations does not seem to pose much of a problem.

On the other hand, as the title of this paper suggests, there is a broad consensus problem in calling this effect, which is due to differences in chemical reaction rates, the 'quantum Zeno effect'. The quantum Zeno effect is generally considered to be based on the 'collapse of the wave function' by quantum measurements and does not include the effect of pure quantum interference of dynamics, as Purtoy (CPL 2010) and Ivanov et al. (JPC 2010) have shown. If they can be interpreted in terms of quantum dynamics between states, there is no special singlet-triplet decoherence or coarse-grained phenomena in radical pairs and no wave packet contraction. The Haberkorn operator is used in precisely such cases, and it has been observed that in the Haberkorn operator the trace of the square of the normalised density matrix is not changed by the Haberkorn operator; some people call the phenomenon in Haberkorn the Zeno effect, but the decay of the state ρ can also be interpreted as a simple broadening of energy due to the decay of states. Of course, in the frequency domain, it can also be interpreted as a broadening of energy due to the imaginary part of the effective Hamiltonian, i.e. the lifetime broadening. Therefore, the 'Zeno effect with collapse of the wave packet due to quantum measurement' in the general interpretation and the 'spread in the energy domain of the probability density of states due to a simple damping term' discussed in the area of spin chemistry are similar but strictly very different. This is also the reason why the term quantum Zeno effect had not been used until the Kominis proposal. However, the rhetoric of personifying of radical pair recombination as if it were an artificial act of observation (e.g. LIF measurements by pulsed laser irradiation), triggered by Kominis' proposal, has resulted in the existence of quantum observations even in natural quantum dynamics under conditions that no one is watching. This extension of the concept makes it appear to the reviewer that the quantum Zeno effect has been extended by anthropomorphism, and that the analysis of the effects of conventional recombination reactions is dressed up as if it were a new concept by the term quantum Zeno effect. In reality, similar reductions of the spin mixing rate have been made in some of the discussions of SNPs and RYDMRs (Koptuyug et al. CPL 1990 etc.) due to the state broadening is a well-documented phenomenon.

The reviewer suggests to use "The effect of asymmetric reaction rate" and not necessary to use "The Zeno effect" or to explain the details of the extension of the concept of "Zeno effect" in spin chemistry field. Especially, "Nature Communications" has a reader in broad range of the scientific field. Therefore, the paradoxical feeling of the Zeno effect should be excluded or careful explanation would be necessary.

In total, the reviewer recommends a major revision of the paper.

(Remarks on code availability)

Version 1:

Reviewer comments:

Reviewer #1

(Remarks to the Author)

The authors have done a very good job in addressing all the comments of all Reviewers, in particular my comments. For the reasons outlined in my first report, the manuscript is of high quality and can be published at Nat. Comm.

Regarding the title change brought about by the comments of Reviewer 3, I have to strongly disagree with the Reviewer's comments, which reflect a limited understanding of the relevant physics. There is nothing anthropomorphic in the discussion of the quantum Zeno effect in the radical-pair mechanism. Comments like this aim at a diversion of the discussion.

It is the relevant interactions of the spin degrees of freedom with the vibrational degrees of freedom entering electron transfer processes that realize a quantum measurement. The quantum Zeno effect is a byproduct of this measurement for the specific parameter regime studied by the authors here and by previous authors. If the Reviewer requires the perspective of the "wave function collapse", that also is involved, as can be seen by studying the physical details of the recombination process.

The title chosen by the authors as a result of the criticism of Reviewer 3, who essentially criticises previous published work instead of the work of the authors of this manuscript, does not serve the general reader, nor the essence of this manuscript. The title involving the phrase "asymmetric recombination rates" sounds obscure, much less general, and much less attractive than the previous title involving the "quantum Zeno" effect. If the authors' scientific judgment asserts that the quantum Zeno effect is at play behind their findings, as is apparent from their comments, with which I agree, they should stand by their judgment and their original title.

The quantum Zeno effect in spin-chemical reactions is now well established, and this manuscript adds quite a bit on top of existing literature. Reviewer 3 is welcome to expose his/her arguments in a paper challenging the genuine presence of the quantum Zeno effect in such reactions, and the community can take it from there.

(Remarks on code availability)

Reviewer #2

(Remarks to the Author)

The authors have responded adequately to all the points I raised and I am satisfied at the final version.

(Remarks on code availability)

Reviewer #3

(Remarks to the Author)

The author's reply to the reviewer's comment was totally convincing and the manuscript was improved to the right direction. I congratulate the publishing this article in Nature communication.

(Remarks on code availability)

Notes on revisions made to manuscript NCOMMS-24-35935 “Magnetosensitivity of Tightly Bound Radical Pairs in Biology is Enabled by Strongly Asymmetric Recombination”

Dear Editor and Reviewers,

We are pleased that our investigation was well received and thank the reviewers for their appraisal and valuable suggestions. We have revised the manuscript, integrating all reviewer suggestions. Below we provide a point-by-point response to their recommendations. The reviewers' statements are printed in blue; our response follows in black, whilst summaries of changes are marked gray.

With our best regards,

Matt C. J. Denton,
Luke D. Smith,
Wenhao Xu,
Jodeci Pugsley,
Amelia Toghil,
Daniel R. Kattnig.

Response to the reviewers

Reviewer 1

Reviewer Statement 1.1 — The authors study in depth the repercussions of the Quantum Zeno effect in the radical-pair mechanism. Specifically, they invoke asymmetric recombination rates in a realistic radical-pair system, namely FAD-superoxide. They find, counterintuitively, that when the radicals are in close proximity, with the dipole interactions and the recombination rates being significantly enhanced, the particular choice of asymmetric recombination rates, for which choice the Quantum Zeno effect is readily manifested, leads to unexpectedly high magneto sensitivity.

Although quantum measurement dynamics, and concomitantly the Quantum Zeno effect in the radical-pair mechanism have been previously discovered and analyzed (Refs 45 and 46), this study goes in far-reaching depth using a realistic radical-pair model, and taking advantage of the Quantum Zeno effect in a much broader biophysical setting involving a full description of dipole interactions, spin-orbit interactions and the resulting spin relaxation.

This study opens up an amazing new window into the possibilities offered by the so-called “non-trivial” quantum effects in biology. The repercussions of this study could well reach beyond the specific problem of magneto sensitivity. The manuscript is extremely well written. Overall, I happily recommend publication.

Response: We thank the reviewer for their assessment of the depth of our work that finds an unexpectedly high magnetosensitivity achieved through the quantum Zeno effect, and their acknowledgement of the wide reach beyond the specific problem of magnetosensitivity.

Reviewer Statement 1.2 — The authors might want to shortly comment on why asymmetric recombination rates are realistic for this particular radical-pair model, and also comment, perhaps in the context of previous work on electron transfer reactions with this system, about the particular numerical values of the recombination rates deep in the Quantum Zeno regime.

Response: We thank the reviewer for their suggestion of discussing why the asymmetric recombination rates in our simulations are realistic for this particular radical-pair model, and their request for more detail on electron transfer reactions in this system.

The activation of molecular oxygen by flavins and flavoproteins has been reviewed extensively [1, 2]. Typical reaction stages are as follows: Reduced flavins react with O_2 in an electron transfer reaction, which yields a caged radical pair of the superoxide anion and the corresponding flavin semiquinone. After spin inversion, the latter is often found to collapse into the flavin C(4a)-hydroperoxide. Pulse radiolysis measurements suggest that this bimolecular association process is fast, while the subsequently decayed to oxidized flavin and hydrogen peroxide at an O_2 independent rate is rate determining [1]. The reaction rate constant of the fast association reaction is central to the suggested reaction mechanism, but difficult to assess as the available kinetic data are limited to the overall reaction.

Massey [1] obtains a rate constant on the order of $k_2 \sim 10^8 \text{ Lmol}^{-1}\text{s}^{-1}$ for the conversion of the flavin semiquinone, FH^\bullet , and $O_2^{\bullet-}$ to the oxidized flavin, F, and H_2O_2 , in free solution. This diffusion-influenced bimolecular rate constant can be used to establish a lower bound of the pertinent association rate constant of FH^\bullet and $O_2^{\bullet-}$ in the encounter complex, which can serve as an order of magnitude estimate for k_S in the suggested model. Assuming a kinetic scheme comprising the diffusive association of the reactants to a precursor complex (rate constant k_d), its dissociation to the free reactants (rate constant k_{-d} , and its actual reaction (e.g. forming the hydroperoxide; rate constant k_r), a steady state analysis gives $k_2^{-1} = (K_A k_r)^{-1} + k_{-d}^{-1}$, where $K_A = k_d/k_{-d}$ is the association constant [3]. For non-interacting reactants, the diffusion rate constant is given as $k_d = 4\pi D\sigma$, where $\sigma \approx 6.5 \text{ \AA}$ is the encounter distance in the reactive complex and D is the mutual diffusion coefficient. Estimating $K_A = 4\pi\sigma^2\delta\sigma$ following Sutin [4], where $\delta\sigma$ is the thickness of the reaction zone, which is frequently postulated to equal 0.8 \AA , and D from the Stokes-Einstein-Sutherland relationship, $D = k_B T / (3\pi\eta\sigma)$, with $\eta \approx 1 \text{ mPa s}$, we obtain $k_r \approx 400 \mu\text{s}^{-1}$. As the estimate neglects the subsequent, rate-limiting decomposition of the C(4a)-hydroperoxide, this value can serve as a lower bound of the actual association rate constant. Thus, the predicted k_r is of the required order of magnitude to induce large magnetosensitivity.

In flavoproteins the reactivity is clearly modulated by the protein environment and the overall reaction with O_2 may be orders of magnitude faster or slower, depending on the specific flavoprotein and its class. The formation of the hydroperoxide is often postulated; alternative reaction pathways exist, such as the formation of flavin-N5-peroxides or pathways avoiding the formation of covalent oxygen adducts altogether. A noteworthy example, for which the flavin hydroperoxide is well established, is the reaction of the neutral flavin radical of glucose oxidase and O_2^- , which reacts to the flavin hydroperoxide with a rate constant of $1 \times 10^9 \text{ L}^{-1}\text{mol}^{-1}\text{s}^{-1}$ [1], suggesting a swift intrinsic association reaction, as required

by the suggested mechanism.

The fast addition of superoxide to the flavin semiquinone is also supported by recent density functional calculations of the oxidation reaction in a bacterial luciferase [5, 6]. Stare concludes that the rate-limiting factor of the oxidation is associated with the change in the spin state of the system (which here is treated explicitly). The minimum energy crossing point of the triplet hydroperoxide-flavin semiquinone complex and the singlet oxidation products was found only 3.4 kcal/mol above the reactants, suggesting a fast reaction using typical well frequencies (on the order of $k_B T/h$).

The reoxidation of the reduced or semi-reduced flavin cofactor has been investigated for the plant cryptochrome AtCry1 [7, 8]. The data are in line with the suggested mechanism and the reactivity of flavins as established for free solutions. Specifically, the reaction appears autocatalytic with fully reduced flavin FADH^- oxidizes more rapidly than the FADH^\bullet radical, in agreement with a reactive FADH^\bullet /superoxide radical pair. The overall reaction rates with O_2 are however small ($\sim 50 \text{ M}^{-1} \text{ s}^{-1}$ for FADH^- in AtCry1). This reflects the fact that the access of oxygen to the flavin binding pocket is sterically hindered, rather than a low intrinsic reactivity. In fact, the reaction of FADH^\bullet and superoxide/hydroperoxide in the binding pocket could not be resolved in these experiments but is expected to occur at a much faster rate [9]. Finally, we note that the requirements on the fast asymmetric recombination can be substantially relaxed in the presence of fast singlet-triplet dephasing (cf. reply to 1.5).

In light of the available data and insights from computational models, it is apparent that the singlet reaction rate constants required to elicit the described effect are feasible even in systems that exhibit slow overall kinetics.

Changes: We have included the above detailed discussion in the Supplementary Information section “Comments on the kinetics of the flavin semiquinone-superoxide reaction” and directed the reader to this within the revised manuscript on page 4 with a concise summary of the key points which reads “The activation of molecular oxygen by flavins and flavoproteins has been reviewed extensively [1, 2], and we anticipate that the required strongly asymmetric recombination rates can arise in radical pairs such as these based on: estimates of the rates across reaction stages [1], on density functional calculations of the fast reaction of the flavin semiquinone with hydroperoxyl radicals [5, 6], and on investigations of the reoxidation of the reduced or semi-reduced flavin cofactor for plant cryptochrome AtCry1 [7, 8]. Whilst the reaction of FADH^\bullet and superoxide/hydroperoxide in the binding pocket could not be resolved in these experiments, it is expected to occur at a fast rate [9]. We provide a detailed discussion on the reaction kinetics of the flavin semiquinone-superoxide reaction in the Supplementary Information (SI) and note that additional singlet-triplet dephasing can further relax the requirement of strongly asymmetric reactivity (see SI Fig. 7).”

Reviewer Statement 1.3 — Along these lines, since the authors consider a triplet-born radical-pair, they could cite the paper “Coherent Triplet Excitation Suppresses the Heading Error of the Avian Compass” at New Journal of Physics 12, 085016 (2010), where it is discussed that triplet-born radical-pairs seem to be advantageous. The authors could then shortly estimate what fraction of the magnetosensitivity they calculate is due to the triplet initial condition, e.g. by repeating the calculation for the opposite values of the recombination rates and starting from the singlet state.

Response: The reviewer points out that triplet-born radical pairs can be advantageous. Indeed, the triplet-born character is essential for the described effect, as only for a triplet-born pair an otherwise

Figure 1: Heatmap plots comparing maximum difference in singlet yield (Δ_{MAX}). Plot A is for a singlet-born RP, whereas plot B is for a triplet-born RP. A clear comparative boost in magnetosensitivity is observed for the triplet-born simulation.

inactive population can be "switched on" for recombination by a reorientation of the geomagnetic field. To explore this within our study we have, as requested, undertaken additional simulations comparing the magnitude of effects when considering both singlet-initialised and triplet-initialised radical pairs. These new simulations demonstrate the clear advantage of triplet-born spin states (see Fig. 1), which confirms the reviewer's point, but here is rooted in the engineered effective Hamiltonian under the quantum Zeno condition.

Changes: We have included a citation to the paper "Coherent Triplet Excitation Suppresses the Heading Error of the Avian Compass" and a concise discussion around the choice of initial states and reversed reaction kinetics. Furthermore, we have added the results of the additional simulations to a figure (Fig. S12) in the SI and directed the reader to this on page 8, where we state: "A detailed analysis (see SI Fig. 5) reveals that large sensitivity ensues if, and only if, r is perpendicular to the dominant hyperfine axis, and if the initial spin state is a triplet rather than singlet (see SI Fig. 12)."

Reviewer Statement 1.4 — In their toy model elaboration on the Quantum Zeno effect, it would be helpful, especially for the general reader, to produce some plots with time in the x-axis, in order to elucidate spin dynamics and connect with the rest of the discussion.

Response: We thank the reviewer for their suggestion to produce plots elucidating the spin dynamics to the reader. We have conducted these for both the singlet and triplet probability for the one-nitrogen radical pair model for different orientations (x, z) of the magnetic field. These demonstrate that, in the presence of singlet recombination only, for the z direction the triplet probability decays from 1 to approximately 0.8 within the first 250 ns but is then "frozen" within the remaining timescale of $2 \mu\text{s}$. In contrast in the x direction the population continues to decay to under 0.4 by $2 \mu\text{s}$ resulting in a significant difference in yields (see Fig. 2).

Changes: We have added a figure with these results to the SI and directed the reader to this at the point the one-nitrogen radical pair model results are introduced on page 8 of the revised manuscript, where we have stated: "For this one-nitrogen radical pair model we have also analysed the time-dependence of the singlet and triplet probabilities (see SI Fig. S11), for which the triplet probability shows a marked difference in x and z directions."

Figure 2: Plots showing the evolution of singlet and triplet probability over a $2\mu\text{s}$ time frame of a one-nitrogen radical pair with $A_{\parallel} = 49$ MHz, the second radical displaced by 4.56 Å along \hat{x} , $k_T = 0$, and $k_S = 3000 \mu\text{s}^{-1}$. The triplet probability flatlines at ~ 0.8 for the case of the \hat{z} -aligned field, but decays to under 0.4 for the \hat{x} -aligned field. Including singlet-triplet dephasing at a rate of $\gamma = 1\text{ns}^{-1}$ induces minimal deviation in the spin dynamics.

Reviewer Statement 1.5 — Since the authors use the Haberkorn recombination operators, It would further be helpful if the authors elaborate on what is the effect of the zero-quantum coherence relaxation term $-(k_S + k_T)(Q_S \cdot \rho + \rho \cdot Q_S - 2Q_S \cdot \rho Q_S)$ discussed in Ref 45 and follow up references. Is this term hidden in Nakajima Zwanzig formalism? If not, would this term alter the main results on spin-relaxation and magneto sensitivity? I do not propose to the authors to get involved in master equation discussions, but S-T relaxation is significant in this strong Quantum Zeno regime and it would be very helpful if it is clearly understood, or if the specific assumptions made in the authors' calculations are clearly defined.

Response: S-T dephasing was implicitly included in the original presentation via exchange and electron-electron dipolar interaction fluctuations treated within the Nakajima Zwanzig formalism in many of the models considered. As different models of the recombination superoperator stipulate different degrees of zero-quantum coherence relaxation and as S-T dephasing will be prominent for a tightly bound radical pair due to both inter-radical interactions and recombination reactions, we have now conducted an explicit study of the effect of their inclusion. We have investigated the effect of S-T dephasing for rates $\gamma = 0, 10, 100$ and $1000 \mu\text{s}^{-1}$. These demonstrate that the dephasing has a negligible effect on the magnitude of the magnetic field sensitivity enhancement which remains at $\Delta_{\text{max}} = 0.19$ on increase of γ . However, increasing γ , the enhancement broadens with respect to susceptible k_S rates, i.e. becomes possible for reaction kinetics that are closer to symmetric (see Fig. 3). We note that similar observations emerged when including electron-electron dipolar and exchange environment interactions within the Nakajima Zwanzig formalism as can be seen in Fig. S6 of the SI. Hence, the effect is robust to or, for small k_S , even further enhanced by the zero-quantum coherence relaxation term.

Changes: We have included the new results within a figure (Fig. S7) in the SI and discussed these effects on page 11 where different noise models are considered. Specifically, we have stated: "Furthermore we have analysed the effect of singlet-triplet dephasing (see Fig. S7), which demonstrates that on increase of the dephasing rate γ up to $1000\mu\text{s}^{-1}$, there is a negligible effect on the sensitivity enhancement, yet there is a broadening with respect to k_S rates permitting the enhancement at reaction kinetics that are closer to symmetric. Similar effects can be seen in the models incorporating ST-dephasing indirectly, i.e. as a result of fluctuating EED and exchange interactions."

Reviewer Minor Statement 1.1 — The authors mention previous work of this reviewer (Refs 45 and 46), but Ref 45 does not deal with magnetosensitivity loss and its recovery in the Quantum

Figure 3: Heatmaps showing simulations for systems where singlet triplet dephasing has been included at varying rates. Increasing the dephasing rate to $\gamma = 1000\mu s^{-1}$, a broadening of the peak sensitivity range, in the direction of slower k_S , illustrates that singlet-triplet dephasing may function as a substitute for recombination if it itself is slower.

Zeno regime. Ref 45 introduces quantum measurement dynamics in the radical-pair mechanism, and the resulting Quantum Zeno effect in the regime of asymmetric recombination rates. This ought to be clearly stated, i.e. citations 45 and 46 ought to be decoupled.

Changes: We have decoupled the references 45 and 46 as suggested.

Reviewer Minor Statement 1.2 — In the caption of Figure 5 there is in the ket T_{Λ} , but in the left figure there is T_{Δ} .

Changes: We thank the reviewer for identifying this typo and have now corrected it.

Reviewer 2

Reviewer Statement 2.1 — One of the major unsolved questions in the field of biological magnetoreception is the mechanism whereby chemical magnetosensing occurs in biological systems, as distinct from ferromagnetite based polarity compass mechanisms. This so-called Radical Pair mechanism involves spin chemical forces which alter reaction rates of biological signaling pathways and thereby result in a directional response to applied magnetic fields. Much physiological and behavioural evidence has implicated flavoproteins, known as cryptochromes, as the likely receptors in this process in the avian system as well as several others. Although it is further demonstrated that biological magnetosensing requires illumination of cryptochrome, the actual magnetosensitive reaction step occurs in the dark, during the process of flavin reoxidation, concomitantly with the formation of the $\text{FADH}^{\bullet}/\text{O}^{\bullet-}$ 2 RP, which has therefore been an obvious candidate for the magnetosensitive radical pair. However, the deleterious effects of spin relaxation arising through the spin rotational mechanisms, and of electron-electron dipolar (EED) coupling of the radical pairs at short inter-radical distances on the magnetosensitive dynamics, severely limit the theoretical response of this system to near-earth strength magnetic fields of the type that birds and biological organisms respond to. Therefore, both the nature of the radical pair and of how they respond in the magnetic field has remained a mystery.

Here the authors report for the first time a possible theoretical solution to this continuing conundrum by investigating the influence that the so-called quantum Zeno effect could exert on a triplet initialised flavin semiquinone-superoxide system such as found in cryptochromes. RPs at close contact may in principle develop remarkable magnetic field sensitivity in weak magnetic fields when immobilised in rigid or semi-rigid matrices, such as being bound in a protein despite being subjected to strong inter-radical coupling, such as the unavoidable electron-electron dipolar interaction. This mechanism solves the dilemma of currently established radical pair mechanisms that require rigid immobilisation while simultaneously having to avoid strong inter-radical coupling – a physical impossibility. Therefore this mechanism not only introduces a new paradigm for exploration of the Radical Pair mechanism in general, but also for the first time provides a theoretical explanation for magnetosensing that actually fits all of the known experimental data for magnetic orientation in the avian model, as well as in others that occur independently of light. Given that this model can also be applied to other proposed magnetosensing systems including in the mitochondria and in the human brain, this theoretical approach can be of wide spread interest to several different disciplines and, even more importantly, provides a clear blueprint for experimental

verification. As lack of a testable, physically sound theoretical mechanism for magnetosensing at near-earth strength has been the single greatest factor holding up progress in the field of biological magnetosensing, I welcome this submission as an important aid to further progress and insight.

Response: We appreciate the reviewer's summary of the ongoing conundrum of cryptochrome magnetosensitivity and biological magnetic field effects and hope that this contribution helps to delineate new possibilities and explanations.

Reviewer Statement 2.2 — These authors use the pigeon Cry4 in their experimental model and implicitly identify this as an avian magnetosensor. However the Cry4 cryptochrome undergoes a very slow flavin redox cycle (half-life for flavin reoxidation is 2 - 6 hours, depending on species) similar to photolyases and distinct from sensory Cry4s such as plant and fly Cry4s. This is incompatible with a signaling (orientation) role as birds orient rapidly (pigeons take off in the homing direction within 5 minutes). Furthermore the avian Cry4 is found in retina in a cell type that contains oil droplets, which renders them opaque to UV light. Since birds orient best in UV, this is incompatible with avian navigation (see eg. Wiltschko et al, *Frontiers in Physiology* 2021; some discussion also in Aguida et al, *Frontiers in Plant Physiology* 2024). In the absence of any supporting experimental evidence for Cry4 as a magnetosensor the authors should state explicitly in the paper that Cry4 was used in their modelling (for convenience?) but is incompatible with current behavioural and histological evidence as having a role in avian magnetosensing.

Response: We thank the reviewer for their request to include more clarification on the use of Cry4 for modelling. We have used Cry4 as it is the only avian cryptochrome for which a x-ray crystallographic structure has so far been resolved, which was a requirement for our molecular dynamics simulations. However, we anticipate the identified mechanism to be generally applicable to various cryptochromes. In support of this, please consider Figs. 4 & 5 and their associated captions, where we demonstrate that as far as the suggested mechanism is concerned, i.e. with view of the FAD binding pocket and residues surrounding the putative superoxide radical, sequence and structural homology lets us expect comparable magnetosensitive reoxidation. Specifically, this broad applicability extends to avian Cry1.

Changes: On the first mention of Cry4 we clarify its usage as "currently the only avian cryptochrome with a resolved structure [10]". We have included the sequence alignment and crystal structure comparison figures within the SI and directed the reader to this in the discussion on page 18 of the revised manuscript, where we have stated:

"We note that the fold and residues surrounding the FAD in its putative magnetosensitive configuration are highly conserved across the cryptochrome family, suggesting that familiar results would be expected in cryptochromes that bind FAD and can be fully reduced. Specifically, avian Cry1 and Cry4 are not expected to show fundamental differences from the point of view of this mechanism. In the SI, we compare different cryptochromes with respect to the structure of homology, including all cryptochromes for which dark state magnetic field effects have been described [11] (see SI Figs. S13 & S14)."

Reviewer Statement 2.3 — The paper would also be improved and have broader appeal if the authors could discuss their results more explicitly with respect to biological effects and known magnetosensing cryptochromes (eg drosophila and plant cry4s, which are proven to have magnetosensing function). For example discuss what kind of changes in reaction rate one might expect at different

```

361                                     4                                     430
1 sp|O77059|CRY1_DROME                FPLIDGAMRQLLAEGWLHHTLANTVATFLTRGGLWQSWEHGLQHLLKYLDADWSVAGNMMWSSSAFE
2 sp|Q5IZC5|CRY1_ERIRU                FPWIDAIMTQLRQEGWIHHLAHAVACFLTRGDLWISWEEGMKVFEELLDADWSVAGSMWLSLSCSSFF
3 sp|Q8QG61|CRY1_CHICK                FPWIDAIMTQLRQEGWIHHLAHAVACFLTRGDLWISWEEGMKVFEELLDADWSVAGSMWLSLSCSSFF
4 sp|Q6ZZY0|CRY1_SYLBO                FPWIDAIMTQLRQEGWIHHLAHAVACFLTRGDLWISWEEGMKVFEELLDADWSVAGSMWLSLSCSSFF
5 tr|A0A219TAG4|A0A219TAG4_COLLI      FPWIDAIMTQLRQEGWIHHLAHAVACFLTRGDLWISWEEGMKVFEELLDADYSINAGNMWLSASAFF
6 tr|A0A2I4SZI9|A0A2I4SZI9_ERIRU     FPWIDAIMTQLRQEGWIHHLAHAVACFLTRGDLWISWEEGMKVFEELLDADYSINAGNMWLSASAFF
7 tr|Q9I912|Q9I912_DANRE             FPWIDAIMTQLRQEGWIHHLAHAVACFLTRGDLWISWEEGMKVFEELLDADYSVAGNMWLSASAFF
8 sp|Q43125|CRY1_ARATH                YPLVDAGMRELWATGWLHDIRIVVVSSFFVK-VLQLPWRWGMKYFDWTLDDADLESALGQYITGTLPD
9 tr|M4WL45|M4WL45_PYPAP              FPWIDAIMTQLRQEGWIHHLAHAVACFLTRGNLWISWEEGMKVFEELLDADWSVAGSMWLSLSCSSFF

```

Figure 4: Above are shown multiple sequence alignment of the central part of the sequence of various cryptochromes, namely, from top to bottom, the cryptochrome of *Drosophila melanogaster*; cryptochrome 1 from *Erithacus rubecula*, *Gallus gallus*, and *Sylvia borin*; cryptochrome 4 of *Columba livia*, *Erithacus rubecula*, and *Danio rerio*; cryptochrome 1 from *Arabidopsis thaliana*; and and cryptochrome 2 from *Pyrrhocoris apterus*. The residues putatively surrounding a superoxide molecule bound in the FAD binding pocket and facing N5 are highlighted, whereby green (yellow) indicates agreement with the residue found at the equivalent position in the sequence of cryptochrome 4 from *Columba livia*, the only avian cryptochrome for which the structure has currently been determined by x-ray crystallography and which was used to underpin the molecular dynamics simulations of superoxide bound to cryptochrome. The grey highlights mark the second tryptophan of the tryptophan triad/tetrad. For the key residues, deviations are seen for the cryptochrome from *D. melanogaster* and cryptochrome 1 from *A. thaliana* only, for which the N5-facing Asn is substituted by Cys and Asp, respectively. These substitutions are known to stabilize the semiquinoid FAD in the anionic and protonated form.

Figure 5: Comparison of the crystal structure of the photolyase homology region of cryptochrome 4 from *Columba livia* (red; ClCry4; PDB ID: 6PU0) and the AlphaFold structure predictions of cryptochrome 1 (blue; ErCry1; UniProt: Q5IZC5) and cryptochrome 4 (green; ErCry4; UniProt: A0A2I4SZI9) from *Erithacus rubecula*: A) overall fold and B) FAD binding pocket highlighting residues that putatively bind superoxide. The fold at the FAD binding site is highly conserved in terms of the overall structure as well as pertinent interaction residues, suggesting that, on the level accessible through mere structure, the proposed model of magnetosensitive re-oxidation applies equally to various cryptochromes, including cryptochrome 1 and cryptochrome 4 from the night migratory European Robin.

Figure 6: Figure showing χ plotted against field strength for a FAD/Z RP. Rates chosen here are $k_S=10^3\mu s^{-1}$, $k_T = 1\mu s^{-1}$

field strengths (rate of flavin reoxidation? Low Level Field? should rates increase or decrease?) with this model. Also, what kinds of other physiological parameters or structural changes in the protein might be affecting these rates (dimerisation? cellular redox potential?) and what kinds of residues/structural features/ amino acid oxygen binding sites might be particularly important for these mechanisms? One intriguing question is whether a scavenging mechanism could occur in combination with the Zeno effect to further amplify the MFE.

Response: The reviewer raises several interesting questions on the nature of reaction rates and field strengths. To elucidate the dependence on the field intensity, we have produced a plot of MFE ($\chi = \overline{\Phi}(B_0)/\overline{\Phi}(B_0 = 0) - 1$) against field strength, where $\overline{\Phi}$ represents the singlet yield averaged over magnetic field directions, for a choice of asymmetric rates $k_T = 1\mu s^{-1}$ and $k_S = 10^3\mu s^{-1}$ found to be auspicious for our toy model neglecting relaxation. We observe that, whilst the enhancement sets in at low field, it persists, and the MFE increases, up to tens of mT (see Fig. 6). Overall, the effect exhibits a low-field effect-like structure, but with vastly enhanced and broadened low-field response extending into the tens of mT region.

In response to reviewer 1's request on the validity of asymmetric recombination rates, we have provided an extensive addition to the SI that justifies the likelihood for asymmetric rates but also explores the contributing factors for this and their effects on reaction rates. In particular, we have considered that the reactivity could be modulated by the protein environment and that the overall reaction with O_2 may be orders of magnitude faster or slower depending on the specific flavoprotein and its class. We have also discussed that reoxidation of reduced or semi-reduced flavin cofactor for plant cryptochrome AtCry1. In the response to this reviewer's first statement, we have also provided a discussion around the specific structural features of importance such as the overall fold and FAD binding pocket highlighting residues that putatively bind superoxide, and have identified the fold as a highly conserved feature in the overall structure as well as pertinent interaction residues.

Furthermore, O_2^- is tightly enclosed by Trp395, Asn391 and Phe379, which slows down its rotational diffusion and hinders the escape of the O_2^- . An arginine (Arg356) could be particularly efficient in this

regard, but the the direct front-on interaction was only seen in trajectory 2, i.e. for the less optimal arrangements, which results following a structural rearrangement of the phosphate binding loop (PBL). The current data thus hint at the structural dynamics of the PBL to be central, as it appears to regulate access to the binding pocket, enforces the optimal radical pair geometry, and gates the product release.

The Scavenging + Zeno proposition also raises an intriguing possibility, but the strength of the present work was in suggesting a mechanism for which it is not required *a priori*. while the scavenging effect could possibly provide an additional boost, currently the nature of the scavenger is not obvious, which somewhat hinders the discussion and integration into the rather clear-cut model discussed here. Likewise, in order to address many of these interesting questions beyond the discussion we have provided, additional comprehensive studies would need to be conducted beyond the present proof of principle work, not only on the theoretical level, but in particular including experimental investigation into physiology, structural biology, and magnetic field effects.

Changes: In addition to the changes for Reviewer Statement 1.1 that elaborate on asymmetric recombination rates, we have added a plot of the MFE against magnetic field strength to the SI (Fig. S10) and directed the reader to this on page 8, where the toy model studies are introduced. We have also elaborated on our discussion of amino acid sidechains of Arg356, Trp395 and Asn391 on page 13 with the above reply. We have also revised and added the following to the conclusion "While the answers to these far-reaching questions will require more concrete and detailed models, ideally motivated from direct experimental observations, the mechanism as suggested solves the previously perceived dilemma of the currently established radical pair mechanism of requiring rigid immobilisation while simultaneously having to avoid strong inter-radical coupling – an impossible balancing act. We hope that the proposed model will lead to a surge in the research of magnetic field effects in reoxidation reactions and provide more clarity about magnetic field effects in biology".

Reviewer 3

Reviewer Statement 3.1 — This paper shows that asymmetry effects in spin-selective chemical reactions play a major role in their magnetic field effects and their anisotropy. To make this more concrete, calculations are performed using a model of the reaction between cryptochrome and superoxide, a candidate molecule for magnetosensitivity in migratory birds.

Although the possibility of a cryptochrome-superoxide model has been demonstrated by some of the authors of this paper, the aspect of the binding between oxygen and cryptochrome has not been experimentally demonstrated except by molecular dynamics simulations, and the formation of radical pairs is completely unknown.

Response: As the reviewer points out, we have conducted molecular dynamics simulations [12] that support the possibility that such a tightly-bound superoxide radical pair could be formed in cryptochrome. This idea and work, however, does not stand in isolation and is in fact motivated by experimental [7, 8] and previous theoretical work [12–14]. The existence of the flavin semiquinone-superoxide radical pair as transient species of the reoxidation of flavins is well established for flavoproteins, as we have already outlined in our response to Reviewer 1's statement 1.2, and for cryptochromes, as described in the works by Müller et al. and van Wilderen et al. [7, 8]. However, what has not been achieved is

linking magnetosensitive phenotype in biology to these processes in a credible, i.e. theoretically sound and comprehensive approach. Previous works were sceptical of such effects as strong electron-electron dipolar coupling and spin rotational relaxation were viewed insurmountable obstacles to significant magnetosensitivity. One of the strengths of this paper is in discovering that such a tightly-bound radical pair can attain significant magnetic field sensitivity provided asymmetric reaction kinetics are operative. However, we agree with the reviewer that much more experimental studies ought to be undertaken to assess magnetic field effects in oxidation reactions. We hope that by providing a credible mechanistic explanation, the field will experience much activity in this direction in the near future.

Changes: We have added the above supporting references to the revised manuscript along with a short comment that more clearly traces prior supporting experimental works in the literature on page 2 that provided a basis for investigating the cryptochrome-based superoxide RP system.

Reviewer Statement 3.2 — There is a broad consensus problem in calling this effect, which is due to differences in chemical reaction rates, the 'quantum Zeno effect'. The quantum Zeno effect is generally considered to be based on the 'collapse of the wave function' by quantum measurements and does not include the effect of pure quantum interference of dynamics, as Purtoy (CPL 2010) and Ivanov et al. (JPC 2010) have shown. If they can be interpreted in terms of quantum dynamics between states, there is no special singlet-triplet decoherence or coarse-grained phenomena in radical pairs and no wave packet contraction. The Haberkorn operator is used in precisely such cases, and it has been observed that in the Haberkorn operator the trace of the square of the normalised density matrix is not changed by the Haberkorn operator; some people call the phenomenon in Haberkorn the Zeno effect, but the decay of the state It can also be interpreted as a simple broadening of energy due to the decay of states. Of course, in the frequency domain, It can also be interpreted as a broadening of energy due to the imaginary part of the effective Hamiltonian, i.e. the lifetime broadening. Therefore, the 'Zeno effect with collapse of the wave packet due to quantum measurement' in the general interpretation and the 'spread in the energy domain of the probability density of states due to a simple damping term' discussed in the area of spin chemistry are similar but strictly very different. This is also the reason why the term quantum Zeno effect had not been used until the Kominis proposed. However, the rhetoric of personifying of radical pair recombination as if it were an artificial act of observation (e.g. LIF measurements by pulsed laser irradiation), triggered by Kominis' proposal, has resulted in the existence of quantum observations even in natural quantum dynamics under conditions that no one is watching.

This extension of the concept makes it appear to the reviewer that the quantum Zeno effect has been extended by anthropomorphism, and that the analysis of the effects of conventional recombination reactions is dressed up as if it were a new concept by the term quantum Zeno effect. In reality, similar reductions of the spin mixing rate have been made in some of the discussions of SNPs and RYDMRs(Koptyug et al. CPL 1990 etc.) due to the state broadening is a well-documented phenomenon.

The reviewer suggests to use "The effect of asymmetric reaction rate" and not necessary to use "The Zeno effect" or to explain the details of the extension of the concept of "Zeno effect" in spin chemistry field. Especially, "Nature Communications" has a reader in broad range of the scientific field. Therefore, the paradoxical feeling of the Zeno effect should be excluded or careful explanation would be necessary. In total, the reviewer recommends a major revision of the paper.

Response: We thank the reviewer for pointing this out and we agree that some readers may see the term 'quantum Zeno effect' and envision a specific scenario where one has a measurement on a pure quantum state that results in wave function 'collapse'. We have thus decided to refrain from using the term in the title to avoid this misinterpretation before the broader scope becomes evident. However, we do not think that complete removal is appropriate, but agree that a careful explanation would make this clearer to the reader.

In short, we do not use the term to imply a quantum measurement paradigm of the reaction event, but use the term "quantum Zeno effect" in the tradition of Hore et al. Ivanov et al. and Kominis et al., whilst assuming that the reaction can be modelled through the Haberkorn reaction operator with additional singlet-triplet dephasing noise. Below we discuss selected statements from the literature suggesting that this usage of term is appropriate and demonstrate that the term is now widely understood in physics beyond the collapse paradigm. However, we agree with the reviewer that a no consensus in calling this effect has been reached, and we have adapted the manuscript to explain this problem.

For one, even in the referenced paper by Ivanov *et al.* [15] refers to the phenomenon as the quantum Zeno effect, pointing out that it is "not an exclusive feature of the approach recently proposed by Kominis" and does not require his "quantum measurement" master equation, but is "present at any rate of the singlet-triplet dephasing in the radical pair, which always accompanies the recombination process." Using the conventional form of the Haberkorn recombination operator, Jones and Hore [16] found that the Zeno effect can be obtained, providing qualitatively similar results as k_T is increased to a generalised quantum measurement approach. They show that, for the singlet population of a singlet-born radical pair, 'As k_T is increased from a value much smaller than the singlet-triplet interconversion frequency (ω), the time-dependence changes from oscillatory, to non-oscillatory decay, to quantum Zeno effect'. Kominis and group have furthermore shown [17] that the observation of the effect is independent of the way recombination is accommodated, and present in his master equation, the Haberkorn, and Hore and Jones models. A common indicator of the quantum Zeno effect is the inverse scaling of the effect size $1/k_S$, which is clearly visible in the density plot in Fig. 2 in the manuscript. Wary about the quantum measurement paradigm that the term originally implied (*vide infra*) and to delineate it from Kominis' approach, some have used "chemical Zeno effect" in recent literature instead, and we follow suit by using the term next to "quantum Zeno effect" in the revised manuscript. However, we do insist that 'quantum Zeno effect' helps to link this study to previous works not only in spin chemistry, but physics broadly (whilst being appropriate beyond the quantum measurement paradigm, as discussed next), thus catering to the broad readership of the journal.

Second, on the concept of measurement we do not think there is an extension of anthropomorphism in usage of the term. The concept of an 'observer' and of 'quantum measurement' has since evolved from the early interpretation of requiring a human characteristic to also allow for more general interpretations such as decoherence-based mechanisms [18]. In the context of open quantum systems the quantum Zeno effect has also been extended to include different manifestations, encompassing not only projective measurements inducing wavepacket collapse, but also interactions with an environment including dissipative processes [19] and generalised quantum operations [20], thereby encompassing a class of phenomena in which a coherent transition is suppressed by an interaction that interrogates whether a transition has occurred or not. Furthermore, it has been shown that the Haberkorn master equation emerges from a first principles description of the radical pair reaction [21], and is thus not just a phenomenological approach incorporating a damping term.

Lastly, whilst state broadening is a well-documented phenomenon to describe spin mixing rate reductions,

it is insufficient here to describe the observations. The lifetime of a state receptive to the geomagnetic field must be on the order of $1 \mu\text{s}$ (such that $\omega\tau \simeq 1$, where ω is the Larmor precession frequency in the geomagnetic field), regardless of the recombination rate constants. However, the lifetime broadening of such a state is only on the order of 1.6 MHz, which is tiny relative to the energy level differences resulting from the electron-electron dipolar interaction ($\sim 1 \text{ GHz}$). Specifically, the magnetosensitivity originates from states with small effective decay rates, which end up in near degeneracy as a result of the recombination and are recoupled by the weak geomagnetic field.

Changes: We followed the reviewers suggestion and changed the title to “Magnetosensitivity of Tightly Bound Radical Pairs in Biology is Enabled by Strongly Asymmetric Recombination”, in order to avoid mentioning the Zeno effect here and risk a loss of generality in its interpretation. Following the suggestion by the reviewer to provide a careful explanation, on the first mention of the “quantum Zeno effect” we have provided the generalised definition and additional references, and revised our wording to avoid the implication of projective measurements inducing complete wavefunction collapse. We also introduce “chemical Zeno effect” (which the original proposers of the name describe as an analogue of the quantum Zeno effect in spin-selective recombination) alongside “quantum Zeno effect” and have included a concise version of the key points of the preceding discussion in the ‘Discussion’ surrounding the origin of the results and Fig. 6 of the manuscript. Specifically, on the first mention on pages 3-4 we state: “The quantum Zeno effect, also referred to as the chemical Zeno effect[22], describes the retardation of state evolution brought about by fast, repeated measurements of a system [23, 24], or more generally an arbitrary quantum operation or quantum semigroup [20]. The manifestation of quantum Zeno dynamics in this case is performed by virtue of the spin-selective recombination reaction of the RP.”. We have also revised a paragraph in the ‘Discussion’ on pages 20-21 of the revised manuscript that now reads: “As demonstrated in the SI, the effects described here result from the quantum Zeno effect, as identified and defined generally [15–17] implying the effect from asymmetric recombination without requiring a strict quantum measurement interpretation [23] (i.e. including environment interactions, dissipative processes [19] and generalised quantum operations [20]) and present in all common treatments of the recombination. This is further supported by our results, where a $1/k_S$ scaling of the imaginary parts of the effective Hamiltonian emerges, e.g. visible in the density plot in Fig. 2, that is indicative of the quantum Zeno effect. The remarkable magnetosensitivity then results, for triplet-born radical pairs, from a fortuitous degeneracy of states, induced by the quantum Zeno effect, that are or are not coupled by the Zeeman interaction to the recombining manifold depending on the direction of the magnetic field. We note that this is not a mere effect of lifetime broadening. The lifetime of a state receptive to the geomagnetic field must be on the order of $1 \mu\text{s}$ (such that $\omega\tau \simeq 1$, where ω is the Larmor precession frequency in the geomagnetic field), regardless of the recombination rate constants. However, The lifetime broadening of such as state is only on the order of 1.6 MHz, which is tiny relative to the energy level differences resulting from the electron-electron dipolar interaction ($\sim 1 \text{ GHz}$) in a tightly bound pair.”

References

1. Massey, V. Activation of molecular oxygen by flavins and flavoproteins. *Journal of Biological Chemistry* **269**, 22459–22462 (1994).

2. Toplak, M., Matthews, A. & Teufel, R. The devil is in the details: The chemical basis and mechanistic versatility of flavoprotein monooxygenases. *Archives of Biochemistry and Biophysics* **698**, 108732 (2021).
3. Mladenova, B., Kattnig, D. R. & Grampp, G. ESR-investigations on the dynamic solvent effects of degenerate electron exchange reactions. Part I: Cyanobenzenes. *Zeitschrift für Physikalische Chemie* **220**, 543–562 (2006).
4. Sutin, N. Theory of electron transfer reactions: insights and hindsight. *Progress in Inorganic Chemistry: An Appreciation of Henry Taube*, 441–498 (1983).
5. Stare, J. Oxidation of Flavin by Molecular Oxygen: Computational Insights into a Possible Radical Mechanism. *ACS omega* (2024).
6. Lawan, N., Tinikul, R., Surawatanawong, P., Mulholland, A. J. & Chaiyen, P. QM/MM molecular modeling reveals mechanism insights into flavin peroxide formation in bacterial luciferase. *Journal of Chemical Information and Modeling* **62**, 399–411 (2022).
7. Müller, P. & Ahmad, M. Light-activated cryptochrome reacts with molecular oxygen to form a flavin–superoxide radical pair consistent with magnetoreception. *Journal of Biological Chemistry* **286**, 21033–21040 (2011).
8. Van Wilderen, L. J., Silkstone, G., Mason, M., van Thor, J. J. & Wilson, M. T. Kinetic studies on the oxidation of semiquinone and hydroquinone forms of Arabidopsis cryptochrome by molecular oxygen. *FEBS Open Bio* **5**, 885–892 (2015).
9. Prabhakar, R., Siegbahn, P. E., Minaev, B. F. & Ågren, H. Activation of triplet dioxygen by glucose oxidase: spin-orbit coupling in the superoxide ion. *The Journal of Physical Chemistry B* **106**, 3742–3750 (2002).
10. Zoltowski, B. D. *et al.* Chemical and structural analysis of a photoactive vertebrate cryptochrome from pigeon. *Proceedings of the National Academy of Sciences* **116**, 19449–19457 (2019).
11. Netušil, R. *et al.* Cryptochrome-dependent magnetoreception in a heteropteran insect continues even after 24h in darkness. *Journal of Experimental Biology* **224**, jeb243000. ISSN: 0022-0949 (Sept. 2021).
12. Deviers, J., Cailliez, F., de la Lande, A. & Kattnig, D. R. Avian cryptochrome 4 binds superoxide. *Computational and Structural Biotechnology Journal* **26**, 11–21 (2024).
13. Salerno, K. M. *et al.* Long-time oxygen and superoxide localization in Arabidopsis thaliana cryptochrome. *Journal of Chemical Information and Modeling* **63**, 6756–6767 (2023).
14. Arthaut, L.-D. *et al.* Blue-light induced accumulation of reactive oxygen species is a consequence of the Drosophila cryptochrome photocycle. *PloS one* **12**, e0171836 (2017).
15. Ivanov, K. L., Petrova, M. V., Lukzen, N. N. & Maeda, K. Consistent Treatment of Spin-Selective Recombination of a Radical Pair Confirms the Haberkorn Approach. *The Journal of Physical Chemistry A* **114**. PMID: 20704353, 9447–9455 (2010).
16. Jones, J. A. & Hore, P. J. Spin-selective reactions of radical pairs act as quantum measurements. *Chemical Physics Letters* **488**, 90–93 (2010).
17. Dellis, A. & Kominis, I. The quantum Zeno effect immunizes the avian compass against the deleterious effects of exchange and dipolar interactions. *Biosystems* **107**, 153–157 (2012).

18. Schlosshauer, M. Decoherence, the measurement problem, and interpretations of quantum mechanics. *Reviews of Modern physics* **76**, 1267–1305 (2004).
19. Burgarth, D., Facchi, P., Nakazato, H., Pascazio, S. & Yuasa, K. Generalized adiabatic theorem and strong-coupling limits. *Quantum* **3**, 152 (2019).
20. Burgarth, D., Facchi, P., Nakazato, H., Pascazio, S. & Yuasa, K. Quantum zeno dynamics from general quantum operations. *Quantum* **4**, 289 (2020).
21. Fay, T. P., Lindoy, L. P. & Manolopoulos, D. E. Spin-selective electron transfer reactions of radical pairs: Beyond the Haberkorn master equation. *The Journal of Chemical Physics* **149** (2018).
22. Berdinskii, V. L. & Yakunin, I. N. Chemical Zeno effect and its manifestations. *Doklady Physical Chemistry* **421**, 163–165. ISSN: 1608-3121 (July 2008).
23. Misra, B. & Sudarshan, E. G. The Zeno’s paradox in quantum theory. *J. Math. Phys.* **18**, 756–763 (1977).
24. Itano, W. M., Heinzen, D. J., Bollinger, J. J. & Wineland, D. J. Quantum zeno effect. *Phys. Rev. A* **41**, 2295 (1990).

REPLY TO REVIEWERS' COMMENTS

We thank the reviewers for their continued appraisal and valuable suggestions. We have revised the manuscript, integrating the remaining suggestion of reviewer #1 and the editorial requests as provided as Author Checklist. Below we provide a point-by-point response to the reviewers' recommendations. Their statements are printed in blue; our response follows in black.

Reviewer #1

The authors have done a very good job in addressing all the comments of all Reviewers, in particular my comments. For the reasons outlined in my first report, the manuscript is of high quality and can be published at Nat. Comm.

Regarding the title change brought about by the comments of Reviewer 3, I have to strongly disagree with the Reviewer's comments, which reflect a limited understanding of the relevant physics. There is nothing anthropomorphic in the discussion of the quantum Zeno effect in the radical-pair mechanism. Comments like this aim at a diversion of the discussion.

It is the relevant interactions of the spin degrees of freedom with the vibrational degrees of freedom entering electron transfer processes that realize a quantum measurement. The quantum Zeno effect is a byproduct of this measurement for the specific parameter regime studied by the authors here and by previous authors. If the Reviewer requires the perspective of the "wave function collapse", that also is involved, as can be seen by studying the physical details of the recombination process.

The title chosen by the authors as a result of the criticism of Reviewer 3, who essentially criticises previous published work instead of the work of the authors of this manuscript, does not serve the general reader, nor the essence of this manuscript. The title involving the phrase "asymmetric recombination rates" sounds obscure, much less general, and much less attractive than the previous title involving the "quantum Zeno" effect. If the authors' scientific judgment asserts that the quantum Zeno effect is at play behind their findings, as is apparent from their comments, with which I agree, they should stand by their judgment and their original title.

The quantum Zeno effect in spin-chemical reactions is now well established, and this manuscript adds quite a bit on top of existing literature. Reviewer 3 is welcome to expose his/her arguments in a paper challenging the genuine presence of the quantum Zeno effect in such reactions, and the community can take it from there.

We thank the reviewer for their insight. We have followed the reviewer's suggestion and changed the title to "Magnetosensitivity of Tightly Bound Radical Pairs in Cryptochrome is Enabled by the Quantum Zeno Effect". This new title combines suggestions of the reviewer with those of the editorial team. We have opted to include "quantum Zeno effect" in the title, because it is more widely recognizable compared to the rather opaque "asymmetric recombinations", which is a terminus specific to the field. We have already provided detailed arguments of why the term is appropriate in the context of our previous reply, which all reviewers have found convincing.

Reviewer #2

The authors have responded adequately to all the points I raised and I am satisfied at the final version.

We thank the reviewer for their efforts and help in improving our manuscript.

Reviewer #3

The author's reply to the reviewer's comment was totally convincing and the manuscript was improved to the right direction. I congratulate the publishing this article in Nature communication.

We are glad that we could convince the reviewer and thank them for their efforts and support.

Editorial Requests

We have implemented the requested changes. Please refer to the provided Author Checklist for details.